# EVA-Gaussian: 3D Gaussian-based Real-time Human Novel View Synthesis under Diverse Camera Settings

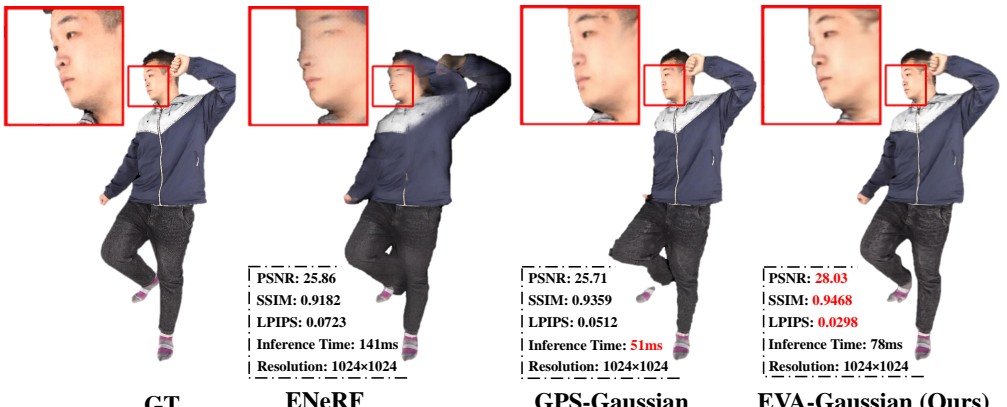

| | PSNR: 25.86 | PSNR: 25.71 | PSNR: **28.03** |
| | SSIM: 0.9182 | SSIM: 0.9359 | SSIM: **0.9468** |
| | LPIPS: 0.0723 | LPIPS: 0.0512 | LPIPS: **0.0298** |
| | Inference Time: 141ms | Inference Time: **51ms** | Inference Time: 78ms |
| | Resolution: 1024×1024 | Resolution: 1024×1024 | Resolution: 1024×1024 |
| **GT** | **ENeRF** | **GPS-Gaussian** | **EVA-Gaussian (Ours)** |

Figure 1: Qualitative comparison of novel view synthesis on the THuman2.0 dataset, with the angle between the stereo views being 72 degree and GT representing the ground truth. We compare our proposed EVA-Gaussian against the state-of-the-art approaches GPS-Gaussian (Zheng et al., 2024) and ENeRF (Lin et al., 2022). The quantitative metrics of PSNR↑, SSIM↑, LPIPS↓, and inference time↓ demonstrate that our proposed method achieves superior reconstruction quality, while enabling real-time reconstruction under sparse viewing conditions and high resolution settings.

## Abstract

The feed-forward based 3D Gaussian Splatting method has demonstrated exceptional capability in real-time human novel view synthesis. However, existing approaches are restricted to dense viewpoint settings, where camera view angles are less than 60 degrees. This limitation constrains their flexibility in free-viewpoint rendering across a wide range of camera view angle discrepancies. To address this limitation, we propose a real-time pipeline named *EVA-Gaussian* for 3D human novel view synthesis across diverse multi-view camera settings. Specifically, we first introduce an Efficient cross-View Attention (EVA) module to accurately estimate the position of each 3D Gaussian from the source images. Then, we integrate the source images with the estimated Gaussian position map to predict the attributes and feature embeddings of the 3D Gaussians. Moreover, we employ a recurrent feature refiner to correct artifacts caused by geometric errors in position estimation and enhance visual fidelity. To further improve synthesis quality, we incorporate a powerful anchor loss function for both 3D Gaussian attributes and human face landmarks. Experimental results on the THuman2.0 and THumansit datasets showcase the superiority of our EVA-Gaussian approach in rendering quality across diverse camera settings. Project page: `https://anonymousiclr2025.github.io/iclr2025/EVA-Gaussian`.

## 1 Introduction

3D reconstruction and novel view synthesis have long been fundamental yet complex tasks in visual data representation and computer vision. Recent advancements in fast 3D reconstruction and

novel view synthesis for humans have shown immense potential in applications such as holographic communication, real-time teaching, and augmented/virtual reality (AR/VR), where time efficiency is critical for user experience and downstream processing. Nonetheless, existing methods either rely on dense input views and precise templates as prior knowledge (Qian et al., 2024; Lei et al., 2024; Hu et al., 2024; Wen et al., 2024; Kocabas et al., 2024; Kwon et al., 2024) or are restricted to specific camera poses (Zheng et al., 2024; Tu et al., 2024). None of these approaches have fully developed a pipeline for real-time human reconstruction under diverse, especially sparse, camera viewpoints, which remains a significant challenge.

In recent years, Neural Radiance Fields (NeRFs) (Mildenhall et al., 2021) have emerged as a promising technique for 3D reconstruction. These models employ neural networks to predict the color and density of sampled 3D points along camera rays and aggregate these predictions to synthesize novel images with high fidelity. Despite their effectiveness, NeRFs suffer from substantial time consumption during both the training and rendering phases. Although various advancements, such as multi-resolution hash encoding (Müller et al., 2022) and feed-forward neural scene prediction (Yu et al., 2021a; Xu et al., 2024), have been made to reduce the time for reconstruction and rendering, the achievable speeds remain insufficient for real-time applications.

More recently, 3D Gaussian Splatting (3DGS) (Kerbl et al., 2023) has been introduced as a solution to this rendering bottleneck. 3DGS utilizes a set of discrete Gaussian representations to model complex 3D scenes and leverages the $\alpha$-blending technique to enable real-time novel view rendering. In the field of 3D human avatar reconstruction, previous works (Qian et al., 2024; Lei et al., 2024; Hu et al., 2024; Wen et al., 2024; Kocabas et al., 2024; Kwon et al., 2024) have employed 3DGS as a representation for humans to achieve animatable full-body human avatar reconstruction. These methods, however, rely on precise human templates as priors, and their training and reconstruction processes can take from minutes to hours, which impedes their use in real-time applications such as holographic communication. While a feed-forward human reconstruction method (Zheng et al., 2024) has achieved fast reconstruction and real-time rendering with stereo inputs, the stereo settings and small angle change between camera viewpoints restrict the overall reconstruction quality under sparse camera settings and lead to sub-optimal performance with more than two input views.

To address these limitations and enable real-time 3D reconstruction of humans using various camera positions and different numbers of cameras, we propose *EVA-Gaussian*, a novel 3D Gaussian-based pipeline for real-time human novel view synthesis. Our method attaches 3D Gaussians to the surface of human body through multi-view depth estimation and aligns their positions closely with point cloud locations. A key innovation of our method is the introduction of an Efficient cross-View Attention (EVA) module for multi-view 3D Gaussian position estimation (see Sec. 4.2). Specifically, we employ a U-Net (Ronneberger et al., 2015) as the backbone and further use dedicated window-embedded cross-view attention to infer multi-view position correspondences. This attention mechanism enables EVA-Gaussian to effectively process multiple inputs from cameras positioned at various viewpoint angles, thereby ensuring robust performance across a wide range of viewing angles, even under extremely sparse camera settings. Besides, we incorporate a Gaussian attribute estimation module that takes the EVA output and the original RGB images as input to estimate the remaining 3D Gaussian attributes (see Sec. 4.3). Furthermore, we embed an additional attribute, referred to as feature, into each Gaussian for further feature splatting and image quality refinement, thereby mitigating the position estimation error introduced by the EVA module (see Sec. 4.4). In addition, we employ an anchor loss to penalize the inconsistency between multi-view face landmarks, which achieves better supervision for human faces (see Sec. 4.5). We conduct extensive experiments on the THuman2.0 (Yu et al., 2021b) and THumanSit (Zhang et al., 2023) datasets. The results, as exemplified in Fig. 1, demonstrate that our proposed EVA-Gaussian outperforms existing feed-forward synthesis approaches in rendering quality, while enabling real-time reconstruction and rendering. Moreover, our approach generalizes well to settings with diverse numbers of cameras and significant changes in camera viewpoint angles. In summary, our main contributions are as follows:

- We propose a novel pipeline for fast feed-forward 3D human reconstruction, called *EVA-Gaussian*, that comprises three main stages: 1) a multi-view 3D Gaussian position estimation stage, 2) a 3D Gaussian attributes estimation stage, and 3) a feature refinement stage.

- We introduce an EVA module to enhance multi-view correspondence retrieval, leading to improved 3D Gaussian position estimation and enhanced novel view synthesis under diverse view numbers and sparse camera settings.

- We employ a recurrent feature refiner that fuses splatted RGB images and feature maps to mitigate geometric artifacts caused by position estimation errors. Moreover, we incorporate an anchor loss that utilizes facial landmarks as anchor points to better supervise Gaussian position estimation, thereby enhancing the quality of synthesized novel view images.

- Extensive experiments on THuman2.0 and THumansit demonstrate the effectiveness and superiority of our proposed pipeline over existing methods in terms of rendered novel view quality and inference speed, especially under sparse camera settings.

## 2 RELATED WORKS

**3DGS-based Human Reconstruction.** 3D Gaussians Splatting has recently emerged as an effective technique for 3D human reconstruction. However, most previous works (Lei et al., 2024; Hu et al., 2024; Wen et al., 2024; Qian et al., 2024; Kocabas et al., 2024; Pan et al., 2024) bind 3D Gaussians to a predefined human mesh model, such as SMPL (Loper et al., 2023) or SMPL-X (Pavlakos et al., 2019). This approach generates 3D Gaussians and human models in a canonical space and then transforms them to match the target human pose using the predefined weights of the human model. This iterative binding process, however, is extremely time-consuming. Moreover, these methods require human templates as inputs at each frame, which incurs extra computational cost and potentially misleads the reconstruction procedure due to the errors in pose estimation. These limitations significantly hinder their applicability in real-world scenarios.

**Fast Generalizable 3D Reconstruction.** In the field of NeRF rendering, pixelNeRF (Yu et al., 2021a) pioneers the approach of predicting features per pixel from a single image in a feed-forward manner for 3D reconstruction. While subsequent works (Chen et al., 2021; Wang et al., 2021; Lin et al., 2022) have followed this feed-forward NeRF pipeline, they still suffer from the extensive time consumption of the NeRF rendering process. Besides, their reconstruction results are often unsatisfactory in sparse camera settings. The introduction of 3DGS has helped mitigate the rendering speed issue of high-quality novel view synthesis methods. Notably, pixelSplat (Charatan et al., 2024) and Splatter Image (Szymanowicz et al., 2024) are the first to combine the feed-forward inference and 3DGS, which predict 3D Gaussian attributes for each pixel and project them back to the 3D space for novel view synthesis in a real-time manner. Nevertheless, these methods still struggle with inaccurate estimation of 3D Gaussian positions. MVSplat (Chen et al., 2024) and MVGaussian (Liu et al., 2024a) address this issue by leveraging cost-volume modules, thereby achieving better novel view image quality. Moreover, latentSplat (Wewer et al., 2024) attaches a latent vector to each 3D Gaussian and uses this vector for novel view refinement through a diffusion decoder and generative loss, which significantly improves image quality on extrapolation views. Despite these advancements, these approaches do not fully exploit prior knowledge about human images and camera settings, which limits their performance on human reconstruction and novel view synthesis. In the field of 3D generation, LGM (Tang et al., 2024) and GSLRM (Zhang et al., 2024) employ a transformer-based feed-forward pipeline for predicting 3D Gaussian attributes from multi-view images in real-world scenarios, but these networks are often too complex for real-time reconstruction. Our work utilizes a carefully designed memory-efficient attention mechanism, thereby enabling generalizable 3D reconstruction at high resolutions with a much lower temporal and computational cost.

The work most closely related to ours is GPS-Gaussian (Zheng et al., 2024), which proposes a stereo matching network for 3D Gaussian position estimation and employs two 3-layer U-Nets to predict 3D Gaussian scales, rotations, and opacities. Although GPS-Gaussian has demonstrated the capability for real-time human reconstruction and novel view synthesis, it suffers from severe distortions under sparse camera settings and mismatch across multiple viewpoints. Subsequent works have attempted to alleviate these issues. For instance, Tele-Aloha (Tu et al., 2024) introduces an image blending and cascaded disparity estimation method for human reconstruction with four input views. However, this approach is tailored to a specific system and struggles to generalize to sparser camera settings. On the other hand, GHG (Kwon et al., 2024) achieves real-time 3D Gaussian-based human novel view synthesis in a feed-forward manner, but it requires additional human template priors, thus inheriting the limitations associated with template-based human reconstruction methods. In contrast, our proposed method eliminates the need for human templates and is specifically designed to generalize effectively across various sparse camera settings.

## 3 PRELIMINARY

**3D Gaussian Splatting (3DGS)** uses a set of 3D Gaussian distributions to represent a 3D sence based on a sequence of multi-view RGB images (Kerbl et al., 2023). Each 3D Gaussian can be mathematically expressed as:

$$G(\boldsymbol{x}) = e^{-\frac{1}{2}(\boldsymbol{x}-\boldsymbol{\mu})^T \boldsymbol{\Sigma}^{-1}(\boldsymbol{x}-\boldsymbol{\mu})}, \tag{1}$$

where $\boldsymbol{\mu} \in \mathbb{R}^3$ and $\boldsymbol{\Sigma} \in \mathbb{R}^{3\times3}$ denote the position vector and the covariance matrix, respectively. The covariance matrix $\boldsymbol{\Sigma}$ is constrained to be semi-positive definite, which is guaranteed by decomposing it into a rotation matrix $\boldsymbol{R} \in \mathbb{R}^{3\times3}$ and a scaling matrix $\boldsymbol{S} \in \mathbb{R}^{3\times3}$ as:

$$\boldsymbol{\Sigma} = \boldsymbol{R}\boldsymbol{S}\boldsymbol{S}^T\boldsymbol{R}^T, \tag{2}$$

where $\boldsymbol{R}$ is further represented by a quaternion $\boldsymbol{q} \in \mathbb{R}^4$, and $\boldsymbol{S}$ is further represented by a scaling vector $\boldsymbol{s} \in \mathbb{R}^3$. In addition, to facilitate rendering from a specific camera perspective, 3DGS leverages a view transformation $\boldsymbol{W}$ to transfer the 3D Gaussians from the world space to the camera space, and employs a projective transformation $\boldsymbol{J}$ to approximate the projection of these 3D Gaussians onto the 2D image plane.

The final rendered color for each pixel is computed using an $\alpha$-blending function that accumulates the contributions of all the projected 3D Gaussians on the pixel:

$$\boldsymbol{c}_{\text{pixel}} = \sum_{i=1}^{N} \boldsymbol{c}_i \alpha_i \prod_{j=1}^{i-1}(1-\alpha_j), \tag{3}$$

where $N$ denotes the number of Gaussians, $\alpha_i$ is the product of the $i$-th projected 2D Gaussian and its optimized opacity value $o_i \in [0,1]$, and $\boldsymbol{c}_i \in \mathbb{R}^3$ denotes the color attribute of the Gaussian, which is represented using spherical harmonics coefficients to enable view-dependent color modeling.

## 4 METHODOLOGY

### 4.1 OVERVIEW

In this paper, we focus on fast human 3D reconstruction and novel view synthesis under diverse camera settings. Our objective is to reconstruct a 3D scene from a set of $n$ sparse-view RGB images $\{\boldsymbol{I}_i\}_{i=1}^n, \boldsymbol{I}_i \in \mathbb{R}^{H\times W\times 3}$, captured from different viewpoints surrounding a human subject, where the angle between any two adjacent camera views is denoted by $\Delta$, and synthesize arbitrary novel view images at any camera position in real time. To achieve this, we propose *EVA-Gaussian*, a method that utilizes deep neural networks and 3D Gaussian Splatting to enhance novel image quality while achieving real-time reconstruction.

Specifically, we employ 3DGS to represent each source image $\boldsymbol{I}_i$ as a set of 3D Gaussians. Each pixel in the foreground corresponds to a unique 3D Gaussian. We use $U_i$ to denote the number of Gaussians for source image $i$. The proposed EVA-Gaussian predicts the positions and attributes of 3D Gaussians in the form of attribute maps $\{\boldsymbol{M}_i\}_{i=1}^n = \{\boldsymbol{P}_i, \boldsymbol{O}_i, \boldsymbol{S}_i, \boldsymbol{Q}_i, \boldsymbol{F}_i\}_{i=1}^n$ from the image set $\{\boldsymbol{I}_i\}_{i=1}^n$, where $\boldsymbol{P}_i, \boldsymbol{O}_i, \boldsymbol{S}_i, \boldsymbol{Q}_i$, and $\boldsymbol{F}_i$ denote the attribute maps for Gaussian positions, opacities, scales, quaternions, and features of source image $i$, respectively. Notably, in the feature map $\boldsymbol{F}_i = \{\boldsymbol{f}_i^u\}_{u=1}^{U_i}$, each element $\boldsymbol{f}_i^u \in \mathbb{R}^{32}$ serves as a new attribute associated with each 3D Gaussian, which will be used later in Sec. 4.4 to remove artifacts caused by geometric errors in $\{\boldsymbol{P}_i\}_{i=1}^n$. Mathematically, the procedure of EVA-Gaussian is expressed as:

$$\{\boldsymbol{M}_i\}_{i=1}^n = \mathcal{D}_{\boldsymbol{\theta}}(\{\boldsymbol{I}_i\}_{i=1}^n), \tag{4}$$

where $\boldsymbol{\theta}$ denotes the parameters of the learnable neural network.

The framework of EVA-Gaussian is depicted in Fig. 2. EVA-Gaussian splits the process of predicting Gaussian maps into three stages. In the first stage, it employs a U-Net architecture with an Efficient cross-View Attention module (EVA) to obtain enhanced multi-view predictions of the 3D Gaussian position maps $\{\boldsymbol{P}_i\}_{i=1}^n$, as elaborated in Sec. 4.2. In the second stage, a Gaussian attribute prediction network, detailed in Sec. 4.3, takes the predicted 3D Gaussian position maps $\{\boldsymbol{P}_i\}_{i=1}^n$

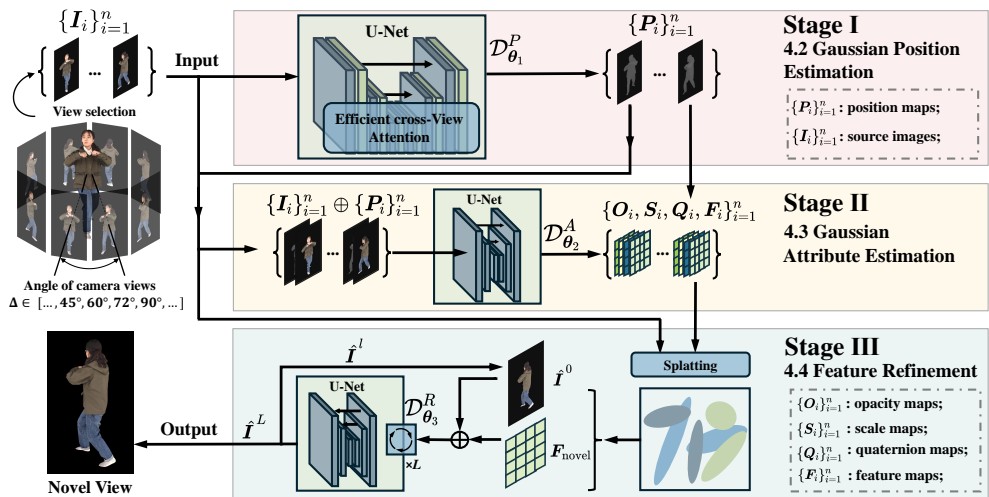

Figure 2: **Framework of EVA-Gaussian**. EVA-Gaussian takes sparse-view images captured around a human subject as input and performs three key stages: (1) estimating the positions of 3D Gaussians, (2) inferring the remaining attributes (i.e., opacities, scales, quaternions, and features) of these Gaussians, and (3) refining the output image in a recurrent manner.

and the original RGB images $\{\boldsymbol{I}_i\}_{i=1}^n$ as input to estimate the remaining attributes of 3D Gaussians. The predicted 3D Gaussians from all source images are then aggregated to render target views using differential rasterization (Kerbl et al., 2023). In the final stage, the rendered RGB image $\hat{\boldsymbol{I}}^0$ and its corresponding feature map $\boldsymbol{F}_{\text{novel}}$ are fused for further refinement using the network described in Sec. 4.4. In addition, an anchor loss is introduced during the training stage to enhance the overall reconstruction quality, as depicted in Sec. 4.5.

## 4.2 GAUSSIAN POSITION ESTIMATION

The variations in depth across the surface of human body may appear minimal. However, these nuances are critically important, particularly in regions such as the face and hands that contain a wealth of semantic information. Even slight inaccuracies in depth estimation within these areas can lead to significant degradation in visual quality and fidelity. This underscores the necessity for precise estimation of 3D Gaussian positions to enable effective and high-fidelity human reconstruction.

To tackle this challenge, we employ a U-Net based architecture $\mathcal{D}_{\boldsymbol{\theta}_1}^P$ to estimate the 3D Gaussian position maps $\{\boldsymbol{P}_i\}_{i=1}^n$ from multi-view images $\{\boldsymbol{I}_i\}_{i=1}^n$, which is expressed as:

$$\{\boldsymbol{P}_i\}_{i=1}^n = \mathcal{D}_{\boldsymbol{\theta}_1}^P(\{\boldsymbol{I}_i\}_{i=1}^n). \tag{5}$$

To ensure accurate depth estimation across diverse camera angles or arbitrary input views, we propose an EVA module, as illustrated in Fig. 3. This module is integrated into the three lowest resolution layers of the U-Net basebone $\mathcal{D}_{\boldsymbol{\theta}_1}^P$ to facilitate the multi-view correspondence retrieval and information exchange. We use $j$ to denote the index of each of these three layers, with $j = -1, j = -2$, and $j = -3$ representing the lowest, the second-lowest, and the third-lowest resolution layers, respectively. EVA takes multiple intermediate image features $\boldsymbol{E}_i^j \in \mathbb{R}^{R^j \times C^j}, \forall i \in \{1, \cdots, n\}, \forall j \in \{-1, -2, -3\}$, as input and outputs the corresponding enhanced image features $\widetilde{\boldsymbol{E}}_i^j$, where $R^j$ and $C^j$ denote the total number of pixels and the channel dimension of each pixel at layer $j$, respectively. Before the execution of attention mechanisms, a learnable positional embedding $\gamma$ is added to the intermediate feature $\boldsymbol{E}_i^j$ to improve the understanding of image coordinates.

In contrast to other feed-forward scene reconstruction methods (Chen et al., 2024; Charatan et al., 2024; Liu et al., 2024a) that apply to a low resolution of $256 \times 256$, our approach aims for high-quality human reconstruction at 1024 resolution. Given that the corresponding pixels from the reference views are located in adjacent locations only under human-centric camera settings, calculating attention scores across a concatenated multi-view image sequence or feature map for the entire image, as adopted by Chen et al. (2024), is highly inefficient. To improve efficiency, EVA computes cross-attention only within a local window, which is shifted by half the window size at each attention

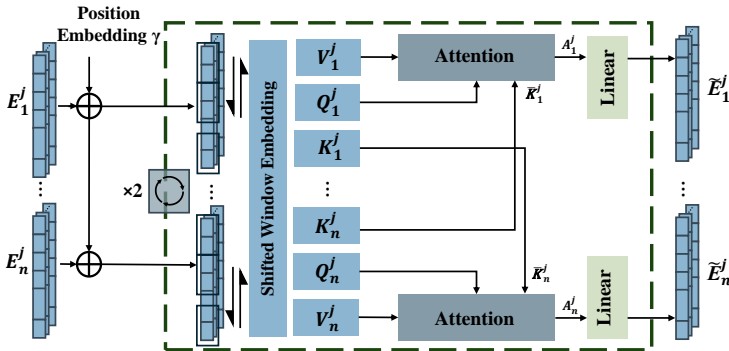

Figure 3: **Efficient cross-View Attention (EVA) module** for 3D Gaussian position estimation. EVA takes multi-view image features as input, embeds them into window patches using a shifted algorithm, and performs cross-view attention between the features from different views.

iteration. This shifted window embedding allows for low computation complexity and better GPU memory utilization, while maintaining high performance.

In the cross-view attention stage, the intermediate feature $E_i^j$ is linearly transformed into query, key, and value matrices, denoted by $Q_i^j$, $K_i^j$, and $V_i^j$, respectively. For each $E_i^j$, we calculate cross-view attention with the key matrices $\overline{K}^j$ fused from all the reference image features, excluding the target feature itself. That is, $\overline{K}_i^j = \mathcal{T}_i^j(K_1^j, \cdots, K_{i-1}^j, K_{i+1}^j, \cdots, K_n^j)$, where the fusion $\mathcal{T}_i^j$ is achieved by using a fully connected neural network. Consequently, each attention map is derived from its associated query, the fused key, and its corresponding value as:

$$A_i^j = \text{softmax}(Q_i^j \overline{K}_i^{jT}/\sqrt{C^j})V_i^j, \tag{6}$$

where $A_i^j, \forall i \in \{1, 2, \cdots, n\}$, denotes the resultant attention output.

Notably, when the scale of each Gaussian is sufficiently small, the 3D Gaussian position of a pixel aligns precisely with its corresponding value on the depth map. A detailed proof of this property is provided in Appendix C. Based on this observation, we train the position estimation network $\mathcal{D}_\theta^{\check{P}}$ to obtain the position maps $\{P_i\}_{i=1}^n$ with the mean squared error (MSE) loss function:

$$\mathcal{L}_{\text{depth}} = ||P_i - P_i^{\text{gt}}||_2, \tag{7}$$

where $P_i^{\text{gt}}$ denotes the ground truth depth map.

### 4.3 GAUSSIAN ATTRIBUTE ESTIMATION

To complete the estimation of 3D Gaussian maps $\{M_i\}_{i=1}^n$, we employ a shallow U-Net $\mathcal{D}_{\theta_2}^A$ to estimate the remaining attributes $O_i$, $S_i$, $Q_i$, $F_i$. This network takes the estimated 3D Gaussian position maps $\{P_i\}_{i=1}^n$ from the first stage in Sec. 4.2 and the original RGB images $\{I_i\}_{i=1}^n$ as input, and outputs the 3D Gaussian attributes $O_i$, $S_i$, $Q_i$, $F_i$, which is expressed as:

$$\{O_i, S_i, Q_i, F_i\}_{i=1}^n = \mathcal{D}_{\theta_2}^A(\{I_i\}_{i=1}^n \oplus \{P_i\}_{i=1}^n). \tag{8}$$

The resulting estimated 3D Gaussian maps $\{M_i\}_{i=1}^n = \{P_i, O_i, S_i, Q_i, F_i\}_{i=1}^n$ are then utilized to render novel views using the process described in Sec. 3. The network $\mathcal{D}_{\theta_2}^A$ is trained by using a combination of MSE loss and structural similarity index measure (SSIM) (Wang et al., 2004) loss between the rendered novel view image $\hat{I}^0$ and the ground truth $I^{\text{gt}}$ as follows:

$$\mathcal{L}_{\text{render}} = ||\hat{I}^0 - I^{\text{gt}}||_2 + \lambda_{\text{render}}(1 - \text{SSIM}(\hat{I}^0, I^{\text{gt}})), \tag{9}$$

where $\lambda_{\text{render}}$ denotes the weighting factor for the SSIM loss.

### 4.4 FEATURE SPLATTING AND REFINEMENT

The 3D Gaussian position maps $P_i$ estimated in Sec. 4.2 inevitably contain some degree of error, which may lead to distortions and artifacts in the rendered RGB images. To mitigate these issues,

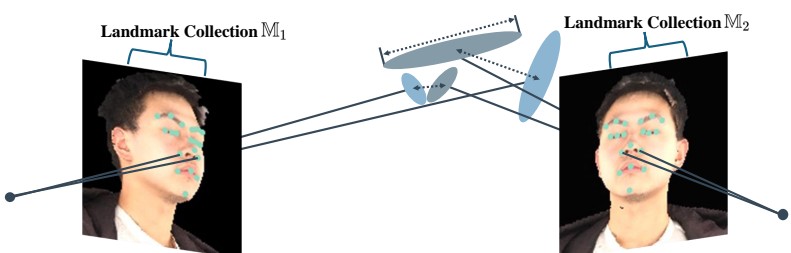

Figure 4: **Attribute regularization.** We regularize the opacities and scales of Gaussians, as well as the position mismatch among the Gaussians in the landmark collection. The optimization of position mismatch terminates when it falls below a specific tolerance.

we propose a post-splatting refinement method to correct the position estimates. Recent studies (Berriel Martins & Civera, 2024) have demonstrated that feature vector representations can capture scene information more effectively than spherical harmonics, resulting in significant improvements in novel view synthesis, particularly in scenarios with limited overlapping views. Inspired by this finding, we incorporate a feature vector, i.e., $\boldsymbol{f}_i^u \in \mathbb{R}^{32}$ mentioned in Sec. 4.1, as an additional attribute for each Gaussian to more precisely capture its spatial characteristics.

During the splatting process, we first aggregate the 3D Gaussians from all source views. Then, the color values of these 3D Gaussians are rendered using Eq. 3. Concurrently, the feature values of the 3D Gaussians are splatted onto the image plane using a modified $\alpha$-blending function as follows:

$$\boldsymbol{f}_{\text{pixel}} = \sum_{j=1}^{N} \boldsymbol{f}_j \alpha_j \prod_{l=1}^{j-1}(1 - \alpha_l), \tag{10}$$

where $\boldsymbol{f}_{\text{pixel}}$ is the feature vector for the corresponding pixel on the feature map of the novel view image $\boldsymbol{F}_{\text{novel}}$, $\boldsymbol{f}_j$ denotes the feature vector for the 3D Gaussian with the $j$-th greatest depth, and $N = \sum_{i=1}^{n} U_i$ is the total number of 3D Gaussians from all source views.

Moreover, we employ a carefully designed recurrent U-Net $\mathcal{D}_{\theta_3}^R$ that takes both the RGB and feature images as input and project them onto the RGB space for the final output through $L$ recurrent loops. This recurrent procedure is expressed as follows:

$$\hat{\boldsymbol{I}}^l = \mathcal{D}_{\theta_3}^R(\hat{\boldsymbol{I}}^{l-1} \oplus \boldsymbol{F}_{\text{novel}}), \hat{\boldsymbol{I}}^l \in \mathbb{R}^{H \times W \times 3}, \boldsymbol{F}_{\text{novel}} \in \mathbb{R}^{H \times W \times 32}, l \in \{1 \cdots L\}. \tag{11}$$

Similar to the Gaussian attribute estimation, the loss function for supervising the final output is a combination of MSE loss and SSIM loss between refined image $\hat{\boldsymbol{I}}^L$ and ground truth $\boldsymbol{I}^{\text{gt}}$ as follows:

$$\mathcal{L}_{\text{refine}} = ||\hat{\boldsymbol{I}}^L - \boldsymbol{I}^{\text{gt}}||_2 + \lambda_{\text{refine}}(1 - \text{SSIM}(\hat{\boldsymbol{I}}^L, \boldsymbol{I}^{\text{gt}})), \tag{12}$$

where $\lambda_{\text{refine}}$ denotes the weighting factor for the SSIM loss.

## 4.5 ATTRIBUTE REGULARIZATION

Since human faces are critical for identification and expression understanding, improving the reconstruction of human faces is much more important than that of other body parts. Previous works like GPS-Gaussian (Zheng et al., 2024) treat the entire human body equally and ignore expressive information contained in human faces. Moreover, they fail to ensure the consistency between depth maps and 3D Gaussian locations, resulting in sub-optimal reconstruction quality in the facial regions.

To address this issue, we propose a regularization term to enhance the overall reconstruction quality. Specifically, our proposed anchor loss regularizes the scales and opacities of Gaussians to ensure consistency between the geometry of predicted depth maps and the 3D Gaussian positions. It also aligns the Gaussians from different views to force their locations to the same landmark. We adopt MediaPipe (Lugaresi et al., 2019) to annotate human facial landmarks and compute the anchor loss to regularize the 3D landmark Gaussian scales, opacities, and positions as follows:

$$\mathcal{L}_{\text{anchor}} = \sum_{i,j \in \mathbb{V}} \sum_{m_i \in \mathbb{M}_i, m_j \in \mathbb{M}_j} \max \left\{ ||\Pi^{-1}(\boldsymbol{m}_i, \boldsymbol{P}_i(\boldsymbol{m}_i)) - \Pi^{-1}(\boldsymbol{m}_j, \boldsymbol{P}_j(\boldsymbol{m}_j))||_2, t \right\}$$

$$+ \lambda_{\text{opacity}} \sum_{i=1}^{N} ||\boldsymbol{O}_i \log(\boldsymbol{O}_i)||_1 + \lambda_{\text{scale}} \sum_{i=1}^{N} ||\boldsymbol{S}_i||_2, \tag{13}$$

Table 1: Comparison with feed-forward 3D reconstruction methods at a resolution of 256×256. Better results are marked in a deeper color.

| $\Delta=45°$ | THuman2.0 | | | THumansit | | | inference time |
|---|---|---|---|---|---|---|---|
| | PSNR↑ | SSIM↑ | LPIPS↓ | PSNR↑ | SSIM↑ | LPIPS↓ | |
| pixelSplat | 25.19 | 0.9156 | 0.0824 | 23.31 | 0.8880 | 0.0954 | 185ms |
| MVSplat | 28.05 | 0.9515 | 0.0346 | 24.97 | 0.9223 | 0.0532 | 70ms |
| MVSGaussian | 26.44 | 0.9706 | 0.0283 | 25.20 | 0.9641 | 0.0297 | 71ms |
| ENeRF | 29.62 | 0.9696 | 0.0238 | 27.06 | 0.9567 | 0.0334 | 136ms |
| GPS-Gaussian | 30.30 | 0.9762 | 0.0224 | 28.02 | 0.9671 | 0.0251 | 40ms |
| EVA-Gaussian | 31.11 | 0.9782 | 0.0198 | 29.16 | 0.9696 | 0.0249 | 55ms |

where $\{\mathbb{M}_i\}_{i=1}^n$ denotes the collection of all landmarks on the 2D image plane, $\mathbb{V}$ denotes the collection of source views, and $\Pi^{-1}$ represents the process of reprojection from 2D image to 3D space.

Since the MediaPipe landmark estimation is not perfectly accurate, we introduce a factor $t$ to control the tolerance for mismatch errors. This tolerance facilitates the optimization by activating the loss only when the landmark distance exceeds $t$. Therefore, this approach optimizes the facial reconstruction loss to a sufficiently low level and avoids being misguided by potential errors in the MediaPipe estimation. This procedure is illustrated in Fig. 4.

By integrating the loss functions in the three stages, i.e., $\mathcal{L}_{\text{depth}}$, $\mathcal{L}_{\text{render}}$, $\mathcal{L}_{\text{refine}}$, and the proposed regularization term $\mathcal{L}_{\text{anchor}}$, the overall training loss of the proposed EVA-Gaussian is given by:

$$\mathcal{L}_{\text{EVA-Gaussian}} = \mathcal{L}_{\text{depth}} + \lambda_1 \mathcal{L}_{\text{render}} + \lambda_2 \mathcal{L}_{\text{refine}} + \lambda_3 \mathcal{L}_{\text{anchor}}, \tag{14}$$

where $\lambda_1$, $\lambda_2$, and $\lambda_3$ are weights used to balance the different loss terms.

Since the 3D Gaussian position and attribute estimation stages can be executed within tens of milliseconds, and feature refinement is lightweight, taking less than ten milliseconds, EVA-Gaussian is capable of rapidly reconstructing 3D human subjects from a collection of RGB images and rendering novel views in a real-time manner.

## 5 EXPERIMENTS

### 5.1 EXPERIMENT SETUP

**Implementation details.** Our EVA-Gaussian is trained on 1024×1024 pixel images across multiple training views using a single NVIDIA A800 GPU for 100K iterations with the AdamW (Loshchilov & Hutter, 2017) optimizer, unless otherwise specified. For the 3D Gaussian position estimation stage, it is first pretrained under the supervision of ground truth depth maps. Baselines are trained using their publicly available code. More implementation details are provided in Appendix A.

**Datasets.** We conduct experiments on two open-source human body datasets: THuman2.0 (Yu et al., 2021b) and THumanSit (Zhang et al., 2023). THuman2.0 contains 526 unique human models with their corresponding SMPL parameters, among which 100 individuals are randomly selected for our evaluation. The THumanSit dataset has a similar structure, containing 72 human models with around 60 poses for each, and we randomly choose 5 individuals with all poses for our evaluation.

**Metrics.** We report results on commonly used metrics: PSNR, SSIM (Wang et al., 2004), and LPIPS (Zhang et al., 2018), computed over the entire image, as some methods may produce artifacts outside the human bounding box (Lin et al., 2022; Zheng et al., 2024). We also include the inference time to demonstrate the real-time reconstruction capability of our method.

### 5.2 STEREO RECONSTRUCTION

**Comparison with state-of-the-art feed-forward reconstruction methods.** We first compare our approach against state-of-the-art (SOTA) feed-forward reconstruction methods, including ENeRF (Lin et al., 2022), pixelSplat (Charatan et al., 2024), MVSplat (Chen et al., 2024), MVSGaussian (Liu et al., 2024b), and GPS-Gaussian (Zheng et al., 2024). All experiments are conducted in a stereo-view setting, where the angle between the two camera views $\Delta = 45°$. The attention modules in the scene reconstruction methods (Charatan et al., 2024; Liu et al., 2024b; Chen et al., 2024) are inefficient in their utilization of GPU memory, limiting their ability to train effectively at a high resolution of 1024×1024. Therefore, we also conduct a fair comparison of all methods at a

Table 2: Comparison of feed-forward human reconstruction methods under different camera angle settings, at a resolution of $1024 \times 1024$. Better results are marked in a deeper color. Notably, GPS-Gaussian fails to work effectively when $\Delta = 90°$, as it is unable to meet its rectification requirement.

| THuman2.0 | $\Delta = 45°$ | | | $\Delta = 60°$ | | | $\Delta = 72°$ | | | $\Delta = 90°$ | | |
| $1024 \times 1024$ | PSNR↑ | SSIM↑ | LPIPS↓ | PSNR↑ | SSIM↑ | LPIPS↓ | PSNR↑ | SSIM↑ | LPIPS↓ | PSNR↑ | SSIM↑ | LPIPS↓ |
|---|---|---|---|---|---|---|---|---|---|---|---|---|
| ENeRF | 27.94 | 0.9573 | 0.0367 | 26.16 | 0.9452 | 0.0516 | 24.61 | 0.9309 | 0.0705 | 22.85 | 0.8990 | 0.1147 |
| GPS-Gaussian | 29.63 | 0.9703 | 0.0174 | 27.36 | 0.9630 | 0.0249 | 24.25 | 0.9519 | 0.0480 | / | / | / |
| EVA-Gaussian | 30.46 | 0.9730 | 0.0178 | 28.29 | 0.9654 | 0.0248 | 27.54 | 0.9614 | 0.0297 | 26.31 | 0.9555 | 0.0391 |
| THumansit | $\Delta = 45°$ | | | $\Delta = 60°$ | | | $\Delta = 72°$ | | | $\Delta = 90°$ | | |
| $1024 \times 1024$ | PSNR↑ | SSIM↑ | LPIPS↓ | PSNR↑ | SSIM↑ | LPIPS↓ | PSNR↑ | SSIM↑ | LPIPS↓ | PSNR↑ | SSIM↑ | LPIPS↓ |
| ENeRF | 25.61 | 0.9397 | 0.0494 | 23.80 | 0.9168 | 0.0745 | 22.48 | 0.8956 | 0.0985 | 21.20 | 0.8571 | 0.1406 |
| GPS-Gaussian | 27.05 | 0.9584 | 0.0227 | 25.19 | 0.9480 | 0.0351 | 21.48 | 0.9276 | 0.0713 | / | / | / |
| EVA-Gaussian | 28.76 | 0.9621 | 0.0236 | 27.38 | 0.9543 | 0.0321 | 26.60 | 0.9498 | 0.0500 | 25.44 | 0.9416 | 0.0512 |

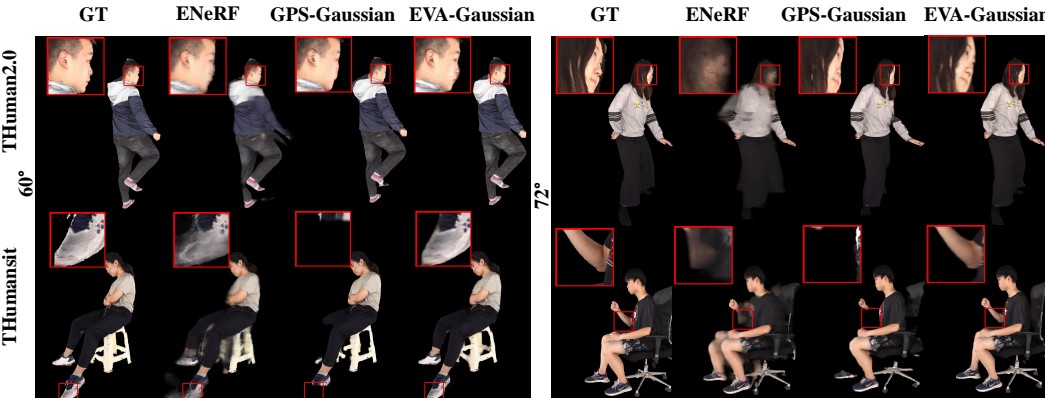

Figure 5: Qualitative comparison on THuman2.0 and THumansit. EVA-Gaussian achieves superior novel view rendering quality under diverse camera settings. Additional visualization results are provided in Appendix B.

resolution of $256 \times 256$. The quantitative results presented in Table 1 demonstrate that EVA-Gaussian achieves the best novel view quality in terms of PSNR, SSIM, and LPIPS, while maintaining the second-fastest inference speed.

**Comparison under diverse angle changes between camera views.** We further evaluate the performance of our method across four different angles between the two camera views, i.e., $\Delta = 45°, 60°, 72°$, and $90°$, at a high resolution of $1024 \times 1024$. As shown in Table 2, our EVA-Gaussian outperforms all baseline methods on all metrics, achieving a maximum PSNR advantage of 5.12 dB. Notably, thanks to our EVA module, EVA-Gaussian remains effective even under extremely sparse camera settings, e.g., $\Delta = 90°$. In contrast, GPS-Gaussian fails to work effectively due to its reliance on stereo rectification. Fig. 5 presents the qualitative results of novel view rendering, where EVA-Gaussian outperforms previous SOTA methods in rendering quality, especially in scenarios with large viewpoint discrepancies.

## 5.3 MULTI-VIEW RECONSTRUCTION

We conduct experiments to compare our method against GPS-Gaussian under multi-view settings. Table 3 presents the quantitative results. Our method demonstrates a significant advantage over the baseline, with a more than 1.5 dB improvement. Notably, the rendering quality of GPS-Gaussian drops significantly due to the mismatch among multiple inferences. In contrast, our method maintains high performance, thanks to the cross-view consistency ensured by our proposed EVA module.

## 5.4 ABLATION STUDY

We conduct a detailed ablation study on THuman2.0 in a stereo-view setting, where the angle between the two views $\Delta = 45°$, as shown in Table 4 and Fig. 6. We gradually incorporate the EVA module, feature refinement module, and anchor loss to evaluate their individual contributions. The absence of the EVA module results in significant degradation across all metrics, and the network

Table 3: Comparison with GPS-Gaussian under different camera number settings. The results in bold represent the best performance. Our EVA-Gaussian achieves SOTA performance across various metrics, primarily due to the multi-view consistency enabled by our proposed EVA module.

| 1024×1024 | THuman2.0 ($\Delta = 45°$) | | | | | | THumansit ($\Delta = 45°$) | | | | | |
| --- | --- | --- | --- | --- | --- | --- | --- | --- | --- | --- | --- | --- |
| | 3 views | | | 4 views | | | 3 views | | | 4 views | | |
| | PSNR↑ | SSIM↑ | LPIPS↓ | PSNR↑ | SSIM↑ | LPIPS↓ | PSNR↑ | SSIM↑ | LPIPS↓ | PSNR↑ | SSIM↑ | LPIPS↓ |
| GPS-Gaussian | 28.74 | 0.9655 | 0.0200 | 28.51 | 0.9636 | 0.0218 | 26.87 | 0.9523 | **0.0243** | 26.50 | 0.9498 | 0.0267 |
| EVA-Gaussian | **30.76** | **0.9722** | **0.0175** | **30.35** | **0.9707** | **0.0189** | **28.64** | **0.9596** | 0.0255 | **28.32** | **0.9582** | **0.0260** |

Table 4: Quantitative results of the ablation study on THuman2.0 in a stereo-view setting, where the angle between the two views $\Delta = 45°$, at a resolution of $1024 \times 1024$.

| 1024×1024 | THuman2.0 ($\Delta = 45°$) | | | |
| --- | --- | --- | --- | --- |
| | w/o EVA module | w/o feature refinement | w/o anchor loss | Full model |
| PSNR↑ | 23.41 | 29.31 | 30.34 | 30.46 |
| SSIM↑ | 0.9380 | 0.9676 | 0.9724 | 0.9730 |
| LPIPS↓ | 0.0659 | 0.0191 | 0.0186 | 0.0178 |

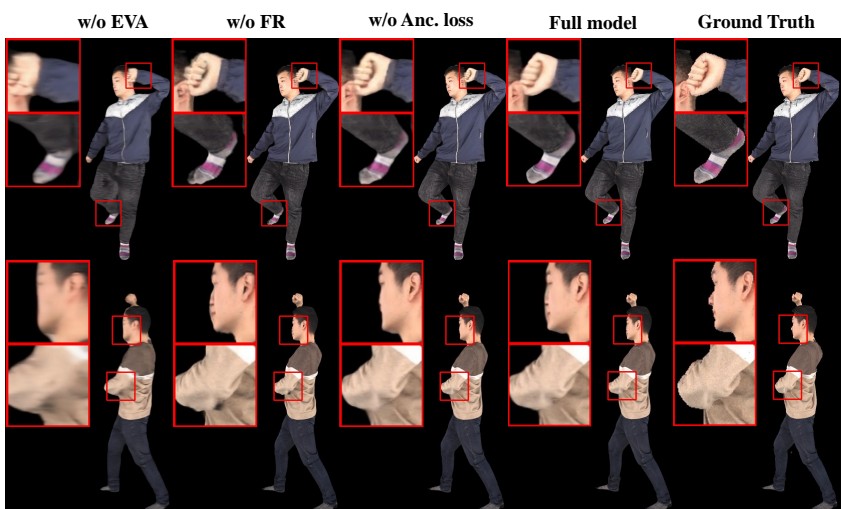

Figure 6: Qualitative visualization results of the ablation study on THuman2.0. Each module shows its effectiveness for a better visual output. The feature refinement (FR) module corrects geometric errors in the initial estimations, and the anchor loss further refines critical areas, such as the face, for generating novel view images with higher fidelity.

struggles to perform multi-view 3D Gaussian geometry prediction. When feature refinement is excluded, visualizations reveal artifacts in critical areas, such as the hands and feet. Moreover, the lack of anchor loss leads to unreliable geometry predictions, particularly in the facial region, which in turn degrades the performance across all metrics, with a notable impact on LPIPS.

## 6 CONCLUSION

In this paper, we introduce EVA-Gaussian, a novel real-time 3D human reconstruction pipeline that employs multi-view attention-based 3D Gaussian position estimation and comprehensive feature refinement. To ensure robust performance, the method is trained using both photometric loss and anchor loss. Quantitative and qualitative evaluations on benchmark datasets demonstrate that our EVA-Gaussian achieves state-of-the-art performance while maintaining a competitive inference speed, particularly under sparse camera settings.

While EVA-Gaussian synthesizes high-fidelity novel views, there remain several areas for improvement. For instance, the attention module can consume substantial GPU memory when processing a large number of input views or high-resolution images. In addition, the naive reprojection of pixels into 3D space may introduce conflicts in overlapping areas, leading to redundancy in the 3D representation. These limitations can be effectively addressed by incorporating RGBD information or developing overlap area detection techniques.

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

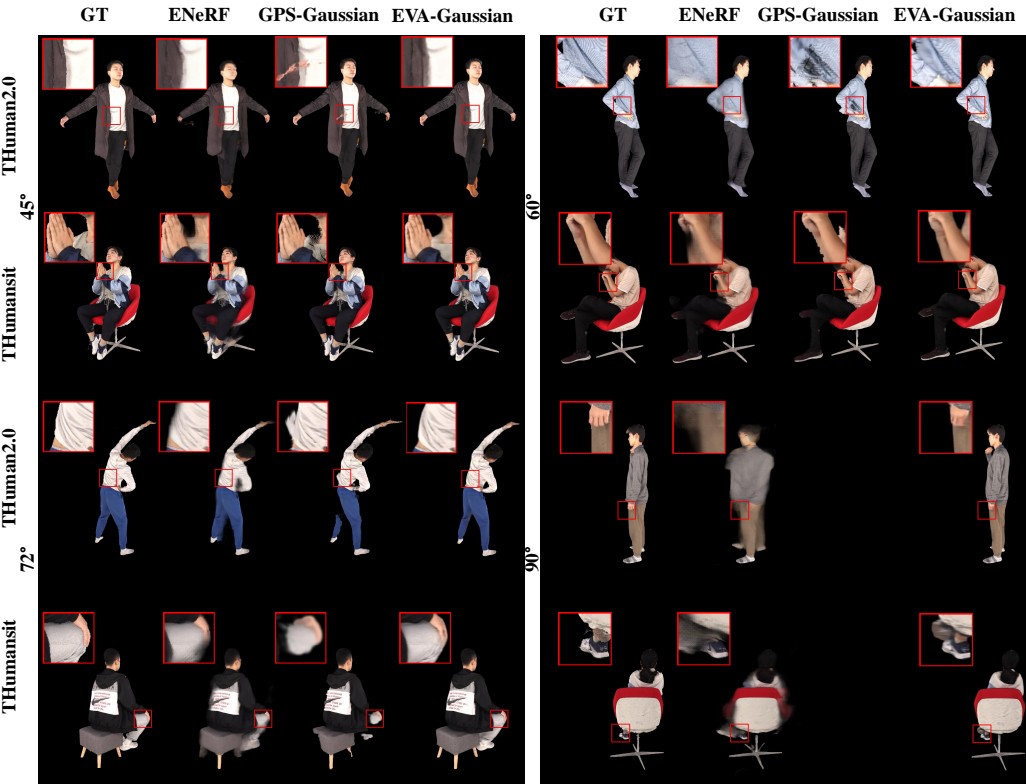

Figure 7: Qualitative comparison on THuman2.0 and THumansit. EVA-Gaussian demonstrates superior novel view rendering quality under diverse camera settings.

## A  MORE IMPLEMENTATION DETAILS

**Network architectures.** Our Gaussian position estimation network $\mathcal{D}_{\theta_1}^P$ utilizes a U-Net as the backbone. The architecture incorporates four stages of $2\times$ down-sampling using average pooling to extract essential feature details. Symmetrically, the network features four stages of $2\times$ up-sampling, achieved through transpose convolutional neural networks. The EVA module is incorporated before the $4\times$, $8\times$, $16\times$ down-sampling and up-sampling blocks. The channel dimension starts at 64 prior to the first down-sampling block, doubling after each down-sampling block and halving after each up-sampling block, which is facilitated by two residual blocks (He et al., 2016). The Gaussian attribute estimation network $\mathcal{D}_{\theta_2}^A$ also employs a U-Net backbone, but it does not include the EVA modules and performs only two stages of $2\times$ down-sampling. The architecture of the feature refiner, $\mathcal{D}_{\theta_3}^R$, mirrors that of $\mathcal{D}_{\theta_2}^A$, but operates in a recurrent manner.

**More training details.** The training hyper-parameters are set as follows: $\lambda_1 = 1, \lambda_2 = 1, \lambda_3 = 10^3, \lambda_{\text{opacity}} = 1, \lambda_{\text{opacity}} = 1, \lambda_{\text{render}} = 0.25$, and $t = 0.05$. The number of recurrent loops $L$ mentioned in Sec. 4.4 is empirically set to $L = 1$ to enhance temporal efficiency. Each training batch contains 2 to 4 source view images, depending on the specific reconstruction task. For instance, the stereo reconstruction task in Sec. 5.2 utilizes 2 source view images. For novel view image supervision, 3 randomly selected views are chosen between each adjacent pair of source views to compute $\mathcal{L}_{\text{refine}}$ and $\mathcal{L}_{\text{render}}$. The learning rate for deep supervised pre-training and overall network training is initialized to 0.0002 and decreases linearly with the number of training epochs.

## B  MORE VISUALIZATION RESULTS

In this section, we present additional visualization results in Fig. 7 to compare our method with SOTA approaches GPS-Gaussian and ENeRF on the Thuman2.0 (Yu et al., 2021b) and Thumansit (Zhang et al., 2023) datasets at a resolution of $1024\times1024$. The results demonstrate that EVA-Gaussian achieves the highest novel view fidelity across various camera viewpoint settings. In con-

trast, GPS-Gaussian struggles to handle the artifacts produced by errors in geometric predictions, while ENeRF generates much more blurry and low-fidelity results compared to both GPS-Gaussian and EVA-Gaussian. Notably, under settings of large viewpoint discrepancy, e.g., $\Delta = 90°$, EVA-Gaussian maintains robust performance, while GPS-Gaussian fails to function effectively in these scenarios.

## C PROOF OF DEPTH EQUALITY

In this section, we prove that for each pixel on the 3D Gaussian maps $\{M_i\}_{i=1}^n$, the rendered depth equals to the predicted 3D Gaussian depth.

We begin by defining the collection for opacity parameters as $o := [o_1, \cdots, o_i, \cdots, o_N] \in \mathbb{R}^N$ of all considered 3D Gaussians and the collection of all 3D Gaussian scaling factors as:

$$\tilde{S} = \begin{bmatrix} s^1, s^2, s^3 \end{bmatrix}^T = \begin{pmatrix} s_1^1 & s_2^1 & \cdots & s_N^1 \\ s_1^2 & s_2^2 & \cdots & s_N^2 \\ s_1^3 & s_2^3 & \cdots & s_N^3 \end{pmatrix}, \tag{15}$$

where $s^1 := [s_1^1, s_2^1, \cdots, s_N^1] \in \mathbb{R}^N$, $s^2 := [s_1^2, s_2^2, \cdots, s_N^2] \in \mathbb{R}^N$ and $s^3 := [s_1^3, s_2^3, \cdots, s_N^3] \in \mathbb{R}^N$. For the 3D Gaussian with the $i$-th greatest depth, the associated scaling matrix is constructed from the corresponding scaling factors as:

$$S_i = \begin{pmatrix} s_i^1 & 0 & 0 \\ 0 & s_i^2 & 0 \\ 0 & 0 & s_i^3 \end{pmatrix}, \tag{16}$$

and its $z$-value is denoted by $z_i$. The rendered depth map is expressed as:

$$D(x) = \sum_{i=1}^N z_i o_i G_i'(x) \prod_{j=1}^{i-1} (1 - o_j G_j'(x))), \tag{17}$$

where $x \in \mathbb{R}^2$ is a variable on the coordinate system of the image plane and $G_i'(x)$ is the 2D Gaussian that corresponds to the 3D Gaussian with the $i$-th greatest depth after splatting.

In the camera's coordinate system, we define a 3D Gaussian as on the reprojected ray of a pixel $x'$, in condition that the center of this 3D Gaussian lies along the ray originating from the camera center and pointing toward the point $[x', 1]$. We use $Z(x')$ to denote the $z$-value of the first 3D Gaussian that appears on this reprojected ray.

Based on the above definitions, we have the following theorem:

**Theorem C.1.** *When the opacity $o$ approaches $1$ and each value in $\tilde{S}$ is sufficiently small, it holds for each pixel $x'$ on the image plane that:*

$$\lim_{\substack{o \to 1 \\ \tilde{S} \to 0^+}} D(x') = Z(x'). \tag{18}$$

Theorem C.1 implies that the $z$-value of the 3D Gaussian at pixel $x$ is equal to the corresponding value on the depth map when the scale of Gaussian is sufficiently small and the opacity approaches 1. To prove Theorem C.1, we introduce the following lemma from the well-known Moore-Osgood Theorem in (Papapantoleon et al., 2023):

**Lemma C.1.** *(Moore-Osgood Theorem) Let $(\Gamma, d_\Gamma)$ be a metric space and $(\gamma_{k,p})_{k,p \in \mathbb{N}}$ be a sequence such that $\gamma_{\infty,p} := \lim_{k \to \infty} \gamma_{k,p}$ exists for every $p \in \mathbb{N}$ and $\gamma_{k,\infty} := \lim_{p \to \infty} \gamma_{k,p}$ exists for every $k \in \mathbb{N}$. If (i) $\lim_{p \to \infty} \sup_{k \in \mathbb{N}} d_\Gamma(\gamma_{k,p}, \gamma_{k,\infty}) = 0$ and (ii) $\lim_{k \to \infty} d_\Gamma(\gamma_{k,p}, \gamma_{\infty,p}) = 0, \forall p \in \mathbb{N}$, then the joint limit $\lim_{k,p \to \infty} \gamma_{n,k}$ exists. In particular, it holds that $\lim_{k,p \to \infty} \gamma_{k,p} = \lim_{p \to \infty} \gamma_{\infty,p} = \lim_{k \to \infty} \gamma_{k,\infty}$.*

Lemma C.1 can be regarded as a special case of Theorem 7.11 from (Rudin et al., 1964). This lemma states that for a doubly-indexed sequence, if the sequence converges uniformly with respect to one

index while converging pointwise with respect to the other index, then the limit of the sequence exists. Moreover, this limit is equivalent to the individual limits obtained by separately considering each index, regardless of the order in which the limiting processes are performed. This result can be extended to continuous multi-variable functions. Specifically, if a continuous function demonstrates uniform convergence with respect to one variable and pointwise convergence with respect to another variable, then the joint limit of the function with respect to both variables can be decomposed into the separate limits with respect to each variable considered independently.

Based on this theoretical foundation, we are now ready to proceed with the proof of Theorem C.1.

*Proof.* When the opacity value $o_i \in \mathbb{R}$ approaches 1 and the scale factor $\boldsymbol{s}_i = [s_i^1, s_i^2, s_i^3] \in \mathbb{R}^3$ is sufficiently small for each Gaussian, the depth value is given by:

$$\lim_{\substack{\boldsymbol{o} \to \mathbf{1} \\ \hat{\boldsymbol{S}} \to \mathbf{0}^+}} D(\boldsymbol{x}') = \lim_{\substack{\boldsymbol{o} \to \mathbf{1} \\ \hat{\boldsymbol{S}} \to \mathbf{0}^+}} \sum_{i=1}^{N} (z_i o_i G_i'(\boldsymbol{x}') \prod_{j=1}^{i-1} (1 - o_j G_j'(\boldsymbol{x}'))) \tag{19}$$

$$= \lim_{\substack{\boldsymbol{o} \to \mathbf{1} \\ \hat{\boldsymbol{S}} \to \mathbf{0}^+}} \sum_{i=1}^{N} (z_i o_i e^{-\frac{1}{2}(\boldsymbol{x}'-\boldsymbol{\mu}_i)^T (\boldsymbol{J}\boldsymbol{W}\boldsymbol{R}_i\boldsymbol{S}_i\boldsymbol{S}_i^T\boldsymbol{R}_i^T\boldsymbol{W}^T\boldsymbol{J}^T)^{-1}(\boldsymbol{x}'-\boldsymbol{\mu}_i)}$$

$$\prod_{j=1}^{i-1} (1 - o_j e^{-\frac{1}{2}(\boldsymbol{x}'-\boldsymbol{\mu}_j)^T (\boldsymbol{J}\boldsymbol{W}\boldsymbol{R}_j\boldsymbol{S}_j\boldsymbol{S}_j^T\boldsymbol{R}_j^T\boldsymbol{W}^T\boldsymbol{J}^T)^{-1}(\boldsymbol{x}'-\boldsymbol{\mu}_j)})) \tag{20}$$

$$\overset{(a)}{=} \lim_{\hat{\boldsymbol{S}} \to \mathbf{0}^+} \lim_{\boldsymbol{o} \to \mathbf{1}} \sum_{i=1}^{N} (z_i o_i e^{-\frac{1}{2}(\boldsymbol{x}'-\boldsymbol{\mu}_i)^T (\boldsymbol{J}\boldsymbol{W}\boldsymbol{R}_i\boldsymbol{S}_i\boldsymbol{S}_i^T\boldsymbol{R}_i^T\boldsymbol{W}^T\boldsymbol{J}^T)^{-1}(\boldsymbol{x}'-\boldsymbol{\mu}_i)}$$

$$\prod_{j=1}^{i-1} (1 - o_j e^{-\frac{1}{2}(\boldsymbol{x}'-\boldsymbol{\mu}_j)^T (\boldsymbol{J}\boldsymbol{W}\boldsymbol{R}_j\boldsymbol{S}_j\boldsymbol{S}_j^T\boldsymbol{R}_j^T\boldsymbol{W}^T\boldsymbol{J}^T)^{-1}(\boldsymbol{x}'-\boldsymbol{\mu}_j)})), \tag{21}$$

where (a) is from Lemma C.1. Specifically, the function $\sum_{i=1}^{N}(z_i o_i G_i'(\boldsymbol{x}) \prod_{j=1}^{i-1}(1 - o_j G_j'(\boldsymbol{x})))$ is continuous with respect to the two variables $\boldsymbol{o}$ and $\boldsymbol{s}$. Besides, it converges uniformly as $\hat{\boldsymbol{S}} \to \mathbf{0}^+$ and as $\boldsymbol{o} \to \mathbf{1}$. This implies that the joint limit of $\boldsymbol{o}$ and $\boldsymbol{s}$ can be decomposed into the separate limits of $\boldsymbol{o}$ and $\boldsymbol{s}$. Thus, we have:

$$\lim_{\hat{\boldsymbol{S}} \to \mathbf{0}^+} \lim_{\boldsymbol{o} \to \mathbf{1}} \sum_{i=1}^{N} (z_i o_i e^{-\frac{1}{2}(\boldsymbol{x}'-\boldsymbol{\mu}_i)^T (\boldsymbol{J}\boldsymbol{W}\boldsymbol{R}_i\boldsymbol{S}_i\boldsymbol{S}_i^T\boldsymbol{R}_i^T\boldsymbol{W}^T\boldsymbol{J}^T)^{-1}(\boldsymbol{x}'-\boldsymbol{\mu}_i)}$$

$$\prod_{j=1}^{i-1} (1 - o_j e^{-\frac{1}{2}(\boldsymbol{x}'-\boldsymbol{\mu}_j)^T (\boldsymbol{J}\boldsymbol{W}\boldsymbol{R}_j\boldsymbol{S}_j\boldsymbol{S}_j^T\boldsymbol{R}_j^T\boldsymbol{W}^T\boldsymbol{J}^T)^{-1}(\boldsymbol{x}'-\boldsymbol{\mu}_j)}))$$

$$= \lim_{\hat{\boldsymbol{S}} \to \mathbf{0}^+} \sum_{i=1}^{N} (\lim_{o_i \to 1} z_i o_i e^{-\frac{1}{2}(\boldsymbol{x}'-\boldsymbol{\mu}_i)^T (\boldsymbol{J}\boldsymbol{W}\boldsymbol{R}_i\boldsymbol{S}_i\boldsymbol{S}_i^T\boldsymbol{R}_i^T\boldsymbol{W}^T\boldsymbol{J}^T)^{-1}(\boldsymbol{x}'-\boldsymbol{\mu}_i)}$$

$$\lim_{(o_j, \cdots, o_{i-1}) \to \mathbf{1}} \prod_{j=1}^{i-1} (1 - o_j e^{-\frac{1}{2}(\boldsymbol{x}'-\boldsymbol{\mu}_j)^T (\boldsymbol{J}\boldsymbol{W}\boldsymbol{R}_j\boldsymbol{S}_j\boldsymbol{S}_j^T\boldsymbol{R}_j^T\boldsymbol{W}^T\boldsymbol{J}^T)^{-1}(\boldsymbol{x}'-\boldsymbol{\mu}_j)})) \tag{22}$$

$$= \lim_{\hat{\boldsymbol{S}} \to \mathbf{0}^+} \sum_{i=1}^{N} (z_i e^{-\frac{1}{2}(\boldsymbol{x}'-\boldsymbol{\mu}_i)^T (\boldsymbol{J}\boldsymbol{W}\boldsymbol{R}_i\boldsymbol{S}_i\boldsymbol{S}_i^T\boldsymbol{R}_i^T\boldsymbol{W}^T\boldsymbol{J}^T)^{-1}(\boldsymbol{x}'-\boldsymbol{\mu}_i)}$$

$$\prod_{j=1}^{i-1} (1 - e^{-\frac{1}{2}(\boldsymbol{x}'-\boldsymbol{\mu}_j)^T (\boldsymbol{J}\boldsymbol{W}\boldsymbol{R}_j\boldsymbol{S}_j\boldsymbol{S}_j^T\boldsymbol{R}_j^T\boldsymbol{W}^T\boldsymbol{J}^T)^{-1}(\boldsymbol{x}'-\boldsymbol{\mu}_j)})). \tag{23}$$

The 3D Gaussians typically assume an ellipsoidal geometric shape. However, when the scaling factors are sufficiently small, the ellipsoid can be approximated as a sphere, such that $s^1 = s^2 = s^3$. As a result, the scaling matrix for the 3D Gaussian with the $i$-th greatest depth becomes:

$$\boldsymbol{S}_i' := \begin{pmatrix} s_i^1 & 0 & 0 \\ 0 & s_i^1 & 0 \\ 0 & 0 & s_i^1 \end{pmatrix}. \tag{24}$$

Consequently, we have:

$$\lim_{\hat{\boldsymbol{S}} \to \boldsymbol{0}^+} \sum_{i=1}^{N} (z_i e^{-\frac{1}{2}(\boldsymbol{x}'-\boldsymbol{\mu}_i)^T (\boldsymbol{JWR}_i\boldsymbol{S}_i\boldsymbol{S}_i^T\boldsymbol{R}_i^T\boldsymbol{W}^T\boldsymbol{J}^T)^{-1}(\boldsymbol{x}'-\boldsymbol{\mu}_i)}$$

$$\prod_{j=1}^{i-1}(1 - e^{-\frac{1}{2}(\boldsymbol{x}'-\boldsymbol{\mu}_j)^T (\boldsymbol{JWR}_j\boldsymbol{S}_j\boldsymbol{S}_j^T\boldsymbol{R}_j^T\boldsymbol{W}^T\boldsymbol{J}^T)^{-1}(\boldsymbol{x}'-\boldsymbol{\mu}_j)}))$$

$$= \lim_{\substack{\hat{\boldsymbol{S}} \to \boldsymbol{0}^+ \\ \boldsymbol{s}^1=\boldsymbol{s}^2=\boldsymbol{s}^3}} \sum_{i=1}^{N} (z_i e^{-\frac{1}{2}(\boldsymbol{x}'-\boldsymbol{\mu}_i)^T (\boldsymbol{JWR}_i\boldsymbol{S}_i\boldsymbol{S}_i^T\boldsymbol{R}_i^T\boldsymbol{W}^T\boldsymbol{J}^T)^{-1}(\boldsymbol{x}'-\boldsymbol{\mu}_i)}$$

$$\prod_{j=1}^{i-1}(1 - e^{-\frac{1}{2}(\boldsymbol{x}'-\boldsymbol{\mu}_j)^T (\boldsymbol{JWR}_j\boldsymbol{S}_j\boldsymbol{S}_j^T\boldsymbol{R}_j^T\boldsymbol{W}^T\boldsymbol{J}^T)^{-1}(\boldsymbol{x}'-\boldsymbol{\mu}_j)})) \tag{25}$$

$$= \lim_{\boldsymbol{s}^1 \to \boldsymbol{0}^+} \sum_{i=1}^{N} (z_i e^{-\frac{1}{2}(\boldsymbol{x}'-\boldsymbol{\mu}_i)^T (\boldsymbol{JWR}_i\boldsymbol{S}_i'\boldsymbol{S}_i'^T\boldsymbol{R}_i^T\boldsymbol{W}^T\boldsymbol{J}^T)^{-1}(\boldsymbol{x}'-\boldsymbol{\mu}_i)}$$

$$\prod_{j=1}^{i-1}(1 - e^{-\frac{1}{2}(\boldsymbol{x}'-\boldsymbol{\mu}_j)^T (\boldsymbol{JWR}_j\boldsymbol{S}_j'\boldsymbol{S}_j'^T\boldsymbol{R}_j^T\boldsymbol{W}^T\boldsymbol{J}^T)^{-1}(\boldsymbol{x}'-\boldsymbol{\mu}_j)})). \tag{26}$$

From (26), we see that when $\boldsymbol{x}' = \boldsymbol{\mu}_i$, it gives that

$$e^{-\frac{1}{2}(\boldsymbol{x}'-\boldsymbol{\mu}_i)^T (\boldsymbol{JWR}_i\boldsymbol{S}_i'\boldsymbol{S}_i'^T\boldsymbol{R}_i^T\boldsymbol{W}^T\boldsymbol{J}^T)^{-1}(\boldsymbol{x}'-\boldsymbol{\mu}_i)} = 1. \tag{27}$$

Otherwise, if $\boldsymbol{x}' \neq \boldsymbol{\mu}_i$, we have

$$\lim_{s_i^1 \to 0^+} e^{-\frac{1}{2}(\boldsymbol{x}'-\boldsymbol{\mu}_i)^T (\boldsymbol{JWR}_i\boldsymbol{S}_i'\boldsymbol{S}_i'^T\boldsymbol{R}_i^T\boldsymbol{W}^T\boldsymbol{J}^T)^{-1}(\boldsymbol{x}'-\boldsymbol{\mu}_i)}$$

$$= \lim_{s_i^1 \to 0^+} e^{-\frac{1}{2(s_i^1)^2}(\boldsymbol{x}'-\boldsymbol{\mu}_i)^T (\boldsymbol{JWR}_i\boldsymbol{R}_i^T\boldsymbol{W}^T\boldsymbol{J}^T)^{-1}(\boldsymbol{x}'-\boldsymbol{\mu}_i)}$$

$$= 0. \tag{28}$$

By combining Eq. 19 – Eq. 28, we have

$$\lim_{\substack{\boldsymbol{o} \to \boldsymbol{1} \\ \hat{\boldsymbol{S}} \to \boldsymbol{0}^+}} D(\boldsymbol{x}') = \sum_{i=1}^{N} (\lim_{s_i^1 \to 0^+} z_i e^{-\frac{1}{2}(\boldsymbol{x}'-\boldsymbol{\mu}_i)^T (\boldsymbol{JWR}_i\boldsymbol{S}_i'\boldsymbol{S}_i'^T\boldsymbol{R}_i^T\boldsymbol{W}^T\boldsymbol{J}^T)^{-1}(\boldsymbol{x}'-\boldsymbol{\mu}_i)}$$

$$\prod_{j=1}^{i-1} \lim_{s_j^1 \to 0^+}(1 - e^{-\frac{1}{2}(\boldsymbol{x}'-\boldsymbol{\mu}_j)^T (\boldsymbol{JWR}_j\boldsymbol{S}_j'\boldsymbol{S}_j'^T\boldsymbol{R}_j^T\boldsymbol{W}^T\boldsymbol{J}^T)^{-1}(\boldsymbol{x}'-\boldsymbol{\mu}_j)})) \tag{29}$$

$$= Z(\boldsymbol{x}'), \tag{30}$$

which completes the proof. $\square$

# D  DETAILS ON EFFICIENT CROSS-VIEW ATTENTION

In this section, we further clarify the motivation of our proposed EVA module, detail its architecture, and analyze its advantages over existing attention mechanisms.

Attention mechanisms that leverage multi-view correspondence across different viewpoints, such as epipolar attention (He et al., 2020), have proven beneficial for downstream 3D tasks like pose estimation. Epipolar attention has demonstrated its effectiveness by performing attention for each pixel only with sampled points along its epipolar line. This approach is based on the principle that a pixel in the source image corresponds to a pixel along the epipolar line in the target image. However, the sampling process and attention calculation in traditional epipolar attention are computationally and temporally intensive, as shown in 5. To mitigate this problem, we propose the EVA module, specifically tailored for the camera settings in feed forward human 3D Gaussian reconstruction,

Table 5: Comparison of the GPU memory usage of different feed forward 3D Gaussian reconstruction methods. Compared to previous feed-forward methods, our EVA-Gaussian method maintains highly competitive GPU memory usage.

| Batch Size=1 | GPU Memory Usage | | | |
|---|---|---|---|---|
| Input Image Resolution | $2 \times 3 \times 128 \times 128$ | $2 \times 3 \times 256 \times 256$ | $2 \times 3 \times 512 \times 512$ | $2 \times 3 \times 1024 \times 1024$ |
| PixelSplat (Epipolar Attention) | 6099 MiB | 13429 MiB | 49598 MiB | Out of Memory |
| MVSplat | 3040 MiB | 6584 MiB | 27082 MiB | Out of Memory |
| GPS-Gaussian | 1909 MiB | 2357 MiB | 4035 MiB | 11215 MiB |
| EVA-Gaussian | 2289 MiB | 3171 MiB | 7185 MiB | 24121 MiB |

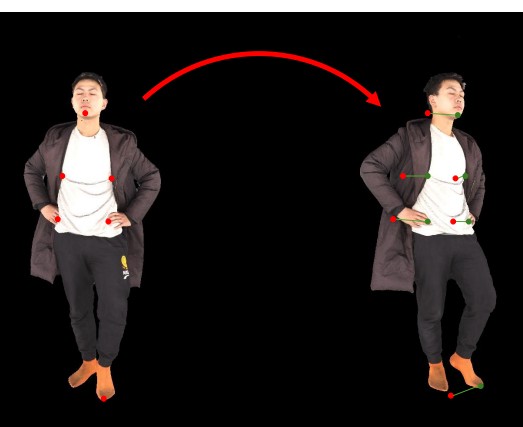

Figure 8: The correspondences between two source view images. The red points in the left view and the green points in the right view are the matching correspondent points. After transferring points from the left view to the right view at the exact position, and connecting them with a line, it is intuitive that the connecting lines are nearly parallel to the x-axis.

where cameras are closely positioned and oriented towards the same point on the human body. In this setting, the corresponding connections between matched pairs align parallel to the x-axis, as depicted in Fig. 8.

In contrast to traditional attention mechanisms, our EVA only computes attention weights for each pixel with nearby pixels along the x-axis. Moreover, we implement this attention mechanism within the deeper layers of the UNet architecture, as shown in Fig. 9, to fuse the dense spatial information from nearby pixels for each pixel. In addition, in order to enable a lager receptive field for each pixel and mitigate the potential loss of multi-view correspondences at the boundaries of local windows at a minimal cost, we perform the attention mechanism twice, with the second iteration using a window shifted by half of its size. Fig. 10 illustrates the key differences between EVA and other attention mechanisms including epipolar attention (He et al., 2020), and the cross attention in LoFTR (Sun et al., 2021). A quantitative comparison between our EVA module and other attention modules in terms of both temporal and computational costs is summarized in Table 6, which demonstrates the efficiency gains achieved through our approach. Our EVA module consumes less than 10% of the time overhead required by MVSplat's attention mechanism.

# E VISUALIZATION FOR DIFFERENT NOVEL VIEW CAMERA SETUPS

In this section, we present additional visualization results under different novel view camera settings.

**Novel View Synthesis under Diverse Viewpoints.** The 3D Gaussians generated under a uniformly placed camera setup with $\Delta = 60°$ illustrate a strong generalization ability to the random novel view camera setup with pitch and yaw ranging from $-25°$ to $+25°$. To be more specific, as shown in Fig. 11, given a pair of input images at $\Delta = 60°$, EVA-Gaussian can effectively infer a human 3D Gaussian model and render it under (A) yaw=0°, pitch=0°, (B) yaw $\in [15°, 25°]$, pitch $\in [15°, 25°]$, (C) yaw=0°, pitch $\in [15°, 25°]$, (D) yaw=0°, pitch $\in [-25°, -15°]$ with promising visual quality.

**Novel View Synthesis Under Higher Resolutions.** We also infer 3D Gaussians using models trained with 1K resolution and render them at both 1K and 2K resolutions. Fig. 12 shows that the 3D

Table 6: Comparison of the temporal and computational efficiency among different attention modules. Notably, the GPU memory consumption of our EVA module does not scale up with window sizes, as the efficient attention algorithm (Shen et al., 2021) is adopted for implementation.

| Input Tensor Size | $2 \times 64 \times 128 \times 128$ | | | $2 \times 64 \times 256 \times 256$ | | | $2 \times 32 \times 256 \times 256$ | | |
|---|---|---|---|---|---|---|---|---|---|
| Method | Time | Params | GPU Memory | Time | Params | GPU Memory | Time | Params | GPU Memory |
| Self-Attention (in MVSplat) | 0.0353s | 0.789M | 3808 MiB | 0.304s | 0.789M | 36290 MiB | 0.263s | 0.198M | 32536 MiB |
| Epipolar Attention (in PixelSplat) | 0.0583s | 5.062M | 15554 MiB | 0.193s | 5.062M | 60562 MiB | 0.169s | 3.194M | 59404 MiB |
| EVA (window size=16) | 0.00722s | 0.0661M | 944 MiB | 0.0177s | 0.0661M | 2200 MiB | 0.0143s | 0.0167M | 1404 MiB |
| EVA (window size=32) | 0.00653s | 0.0661M | 944 MiB | 0.0149s | 0.0661M | 2192 MiB | 0.0116s | 0.0167M | 1404 MiB |
| EVA (window size=64) | 0.00630s | 0.0661M | 944 MiB | 0.0139s | 0.0661M | 2192 MiB | 0.0106s | 0.0167M | 1404 MiB |
| EVA (window size=256) | 0.00656s | 0.0661M | 944 MiB | 0.0234s | 0.0661M | 2192 MiB | 0.0167s | 0.0167M | 1404 MiB |

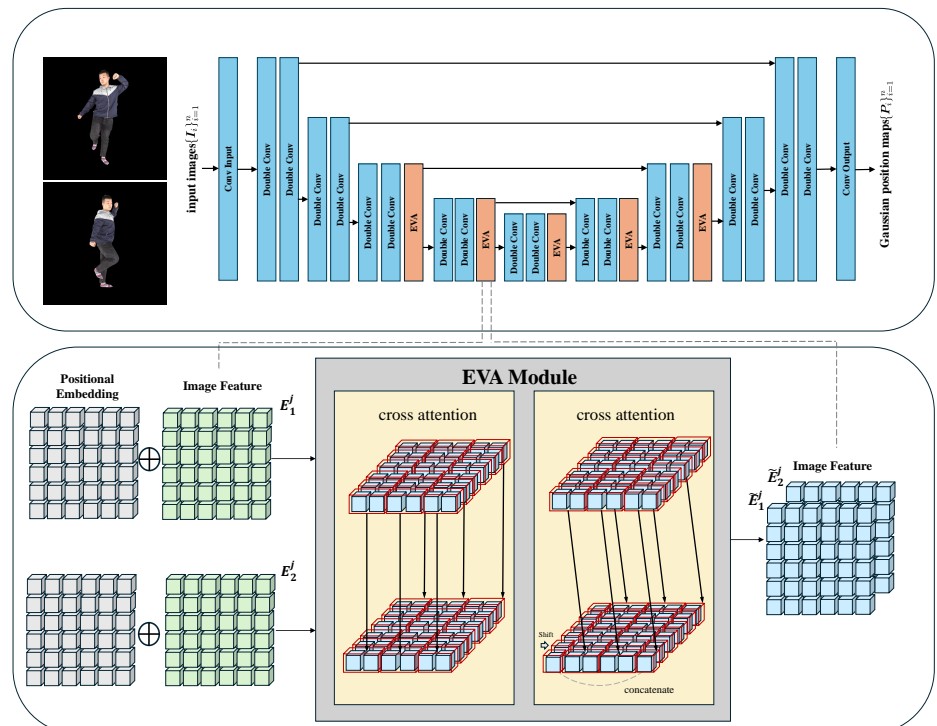

Figure 9: Detailed model structure for the Gaussian Position estimation network $\mathcal{D}_{\boldsymbol{\theta}_1}^P$ in Sec. 4.2. EVA is implemented in the middle layers of the UNet.

Gaussians generated by GPS-Gaussian and EVA-Gaussian can be rendered at 2K resolution. However, artifacts such as holes and incompleteness exist in the 2K renderings of GPS-Gaussian, likely due to the lack of supervision from 2K novel view images. In contrast, although EVA-Gaussian is also not supervised by 2K novel view images, it exhibits greater robustness across different rendering resolutions, which mainly benefits from the stable performance of our feature refinement module.

## F    CROSS-DOMAIN EVALUATION

In this section, we evaluate the cross-domain capabilities of EVA-Gaussian. We first perform evaluation on THumansit dataset with a model trained on THuman2.0 dataset, and subsequently perform evaluation on THuman2.0 dataset with models trained on THumansit dataset. The camera view angles are consistently set to $\Delta = 60°$.

Table 7 presents a comparison on the cross-domain generalization abilities between EVA-Gaussian and GPS-Gaussian. Given that THumansit contains significantly more human models (over 4,000) compared to THuman2.0 (526 models), the performance of both EVA-Gaussian and GPS-Gaussian trained on THumansit perform robustly when evaluated on the THuman2.0 dataset.

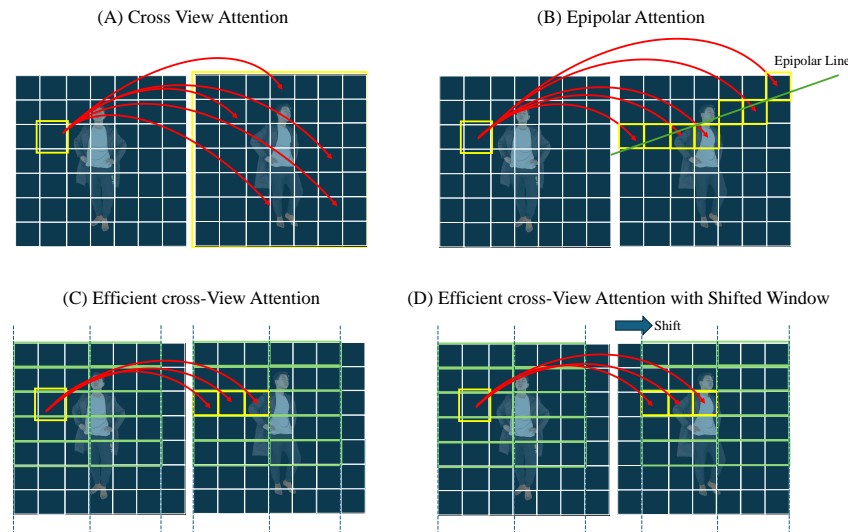

Figure 10: Comparison between different attention mechanisms. (A) is a cross attention mechanism adopted by the area of feature matching, e.g. LoFTR Sun et al. (2021) (B) is epipolar attention from epipolar transformer He et al. (2020), (C) and (D) are the proposed Efficient cross-View Attention at different window embedding stage. EVA only does attention with the most relevant pixels, thus greatly reduces the computational and temporal overhead.

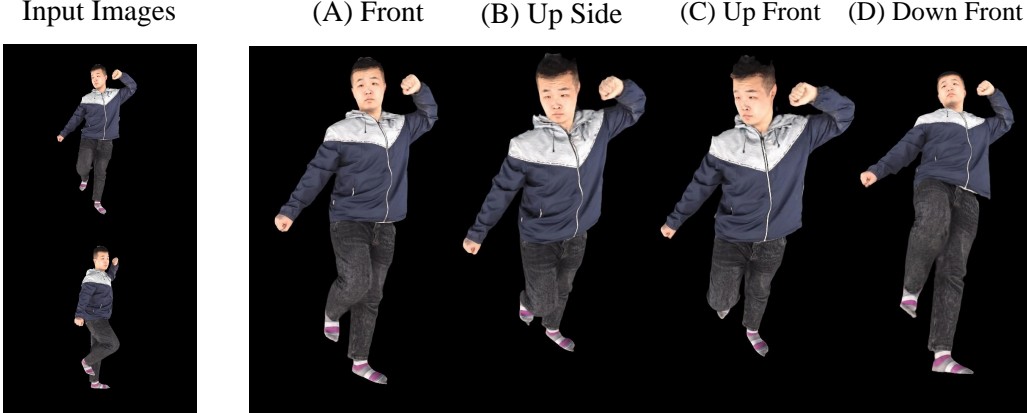

Figure 11: Qualitative results under diverse novel view camera settings demonstrate that the 3D Gaussian model inferred by EVA-Gaussian consistently achieves high-quality novel view renderings across various random camera configurations.

Furthermore, when trained on THumansit dataset and evaluated on THuman2.0 dataset, EVA-Gaussian shows a greater performance improvement (+2.11 dB in PSNR) compared to GPS-Gaussian (+1.97 dB in PSNR). This is attributed to the strong data processing ability of attention modules in EVA-Gaussian, allowing EVA-Gaussian to maintain consistent robustness when provided with sufficient data. This is further illustrated by the evaluation on THumansit, where models trained on THuman2.0 experience a performance decline due to limited data availability; however, EVA-Gaussian still outperforms GPS-Gaussian, achieving a performance gain of 0.41 dB in PSNR under these conditions. In addition, to demonstrate EVA-Gaussian's strong generalization ability across datasets, we present visualization results in Fig. 13, which shows that EVA-Gaussian do not suffer greatly from the out-of-domain problem, since we have explicitly introduced inductive bias to our EVA module.

In conclusion, EVA-Gaussian demonstrates effective generalization ability to cross-domain datasets and exhibits exceptional robustness when a sufficient amount of data is available. Moreover, EVA-Gaussian consistently achieves superior performance on the out-of-domain data.

Input Images    (A) Ground Truth   (B) EVA-Gaussian (C) GPS-Gaussian

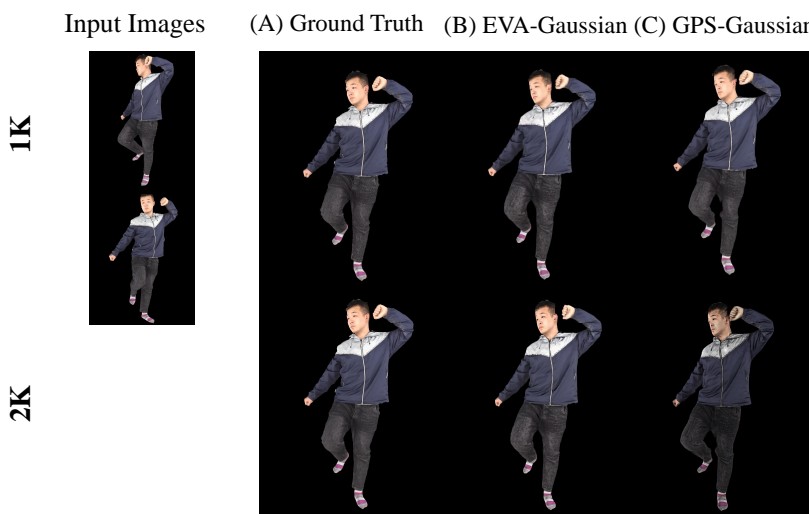

Figure 12: Qualitative rendered results across different resolutions, compared with GPS-Gaussian. Both models are trained with 1K resolution images. There exist incomplete artifacts in the 2K rendering of GPS-Gaussian. In contrast, EVA-Gaussian does not exhibit this issue and produces high-quality rendering results at 2K resolution.

THuman2.0 → THumansit       THumansit → THuman2.0

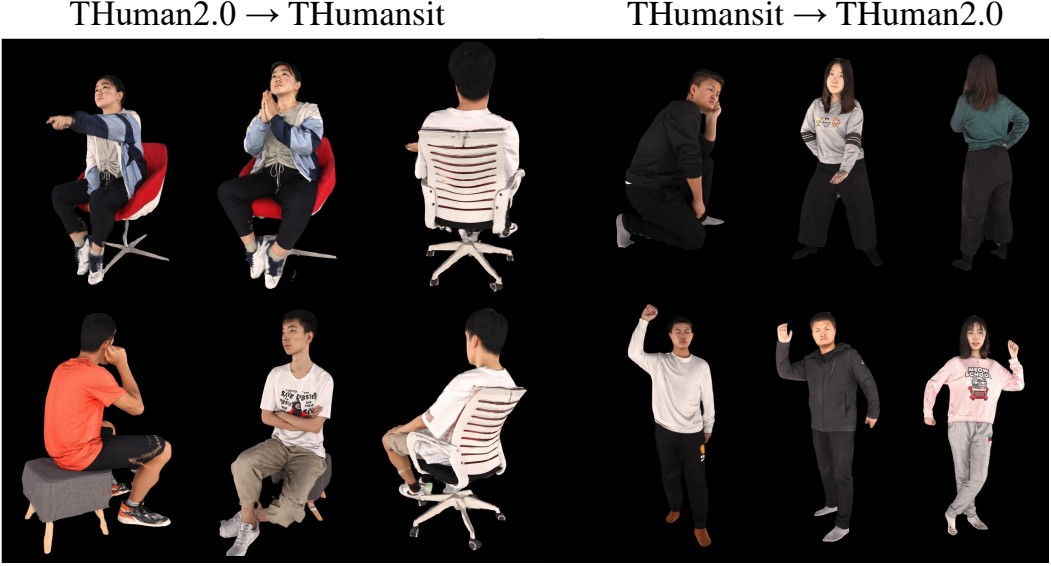

Figure 13: Visualization of cross-domain evaluation results for EVA-Gaussian. The left side displays the rendered results generated by EVA-Gaussian trained on the THuman2.0 dataset and evaluated on the THumansit dataset, while the right side shows the rendered results from EVA-Gaussian trained on the THumansit dataset and evaluated on the THuman2.0 dataset.

Table 7: Quantitative results of cross-domain validations, compared with GPS-Gaussian. It demonstrates that our EVA-Gaussian consistently outperforms GPS-Gaussian when evaluated on various training and evaluation datasets.

| Method | THumansit → THuman2.0 | | | THuman2.0 → THumansit | | |
|---|---|---|---|---|---|---|
| | PSNR↑ | SSIM↑ | LPIPS↓ | PSNR↑ | SSIM↑ | LPIPS↓ |
| GPS-Gaussian | 29.33 | 0.9733 | 0.0325 | 20.86 | 0.9243 | 0.0872 |
| EVA-Gaussian | 30.40 | 0.9751 | 0.0321 | 21.27 | 0.9275 | 0.0876 |

Input Images Δ=90°     (A) Ground Truth     (B) EVA-Gaussian     (C) GPS-Gaussian

Figure 14: Visualization of EVA-Gaussian on real-world data. Minor artifacts on the human boundary mainly arise from the noisy human mask. Notably, GPS-Gaussian cannot generate reasonable outcome under this camera setting.

## G   REAL-WORLD DATA EVALUATION

In this section, we evaluate our model on on real-world data, the HuMMan Cai et al. (2022) dataset, which is a real-world dataset captured with RGB cameras at 1K resolution.

We select images from the front two cameras (ID: 1 and ID: 9) as inputs, infer the 3D Gaussians through EVA-Gaussian model and render novel view on the viewpoint of ID:0. The visualization results, as illustrated in Fig. 14, demonstrate that EVA-Gaussian produces high-quality novel view images in real-world settings.

It is important to note that, due to the sparse input view angles of only $90°$, GPS-Gaussian is unable to generate reasonable outcomes.

