# OpenReview forum: "EVA-Gaussian: 3D Gaussian-based Real-time Human Novel View Synthesis under Diverse Camera Settings"
_ICLR.cc/2025/Conference — Submitted to ICLR 2025_

### Official Review · Reviewer_bSU5 · 2024-10-27

**Soundness:** 3
**Presentation:** 3
**Contribution:** 3
**Rating:** 6
**Confidence:** 3

**Summary:**

This paper proposes a novel EVA-Gaussian that uses multi-view images to synthesize human novel views. It introduces multi-view transformer into the field of human NVS and makes improvements to improve efficiency. It also proposes two concise feature refinement and anchor loss to enhance detail performance. Experiments on two prevailing datasets can well validate the claimed contributions and state-of-the-art performance compared to existing methods.

**Strengths:**

- Performance is good both quantitatively and qualitatively. The results are promising.
- This work can work with significant changes in camera viewpoint angles.
- The paper is easy to follow.

**Weaknesses:**

- The cross-domain generalizability should be further discussed. The paper seems to only conduct in-domain tests. As a generalizable method, it is necessary to evaluate its cross-domain generalizability on different datasets or data captured in the real world.
- Resource cost. As multi-view images are aggregated by a unified attention module, there may be a higher GPU memory cost than previous works. Need more reported results, explanations, and discussion about the consumption.
- Typos. In figure 2, "4.1", "4.2", "4.3" should be "4.2", "4.3", "4.4" to align with the sections.

**Questions:**

Refer to weaknesses.

---

> ### Author Response · Authors · 2024-11-17
> **Response 1/2 to Reviewer bSU5 (Q1-3)**
>
> # Dear Reviewer bSU5,
>
> We sincerely thank the reviewer for the constructive comments. We hope the following responses well address the concerns.
>
> **Q.1** **The cross-domain generalizability should be further discussed. The paper seems to only conduct in-domain tests. As a generalizable method, it is necessary to evaluate its cross-domain generalizability on different datasets or data captured in the real world.**
>
> **A.1** Thank you for this comment. We totally agree that cross-domain generalizability is important for a generalizable reconstruction method. In response to your comment, we have conducted additional experiments and analyses to evaluate the cross-domain generalization capabilities of EVA-Gaussian. Specifically, we compare EVA-Gaussian with GPS-Gaussian across multiple diverse datasets, as detailed in the revised Appendix. Moreover, we have included visualization results of EVA-Gaussian applied to real-world data to demonstrate its practical applicability. Our experimental results suggest that EVA-Gaussian generalizes well across different domains and enhances robustness when provided with sufficient training data.
>
> **Q.2** **Resource cost. As multi-view images are aggregated by a unified attention module, there may be a higher GPU memory cost than previous works. Need more reported results, explanations, and discussion about the consumption.**
>
> **A.2** Thank you for this comment. We admit that the use of attention mechanisms does lead to increased GPU memory consumption, as discussed in our paper (lines 535-539) and illustrated in Table 1 below. However, compared to previous feed-forward methods such as PixelSplat, MVSplat, and GPS-Gaussian, our EVA-Gaussian method maintains highly competitive GPU memory efficiency. Specifically, EVA-Gaussian ranks as the second-lowest in GPU memory demands among these methods across all tested image resolutions. GPS-Gaussian is more memory-efficient because it utilizes stereo matching instead of attention mechanisms for estimating 3D Gaussian positions. This approach, however, results in significant distortion and incompleteness when handling large variations in camera view angles, as detailed in Sec. 5.2 and Sec. 5.3 of our paper.
>
> Moreover, as presented in Table 2 below, EVA-Gaussian outperforms all attention-based methods, including PixelSplat and MVSplat, by requiring the least GPU memory cost and achieving the fastest inference time. This efficiency underscores the effectiveness of our approach in balancing memory consumption with performance.
>
> **Q.3** **Typos. In figure 2, "4.1", "4.2", "4.3" should be "4.2", "4.3", "4.4" to align with the sections.**
>
> **A.3** Thank you very much for your careful review. We have corrected the typos in the revised paper.

---

> ### Author Response · Authors · 2024-11-17
> **Response 2/2 to Reviewer bSU5 (Tables)**
>
> **Table 1** Comparison of the temporal and computational efficiency among different different feed-forward 3D Gaussian reconstruction methods.
>
> | Batchsize=1                           |                       |     GPU Memory Usage      ||                      |
> |---------------------------------------|-----------------------|----------|------------------|---------------------|
> | Input Image Resolution                | 2 * 3 * 128 * 128|2 * 3 * 256 * 256 |2 * 3 * 512 * 512 |2 * 3 * 1024 * 1024 |
> | PixelSplat                            | 6099 MiB         | 13429 MiB        | 49598 MiB        | Out of Memory    |
> | MVSplat                               | 3040 MiB         | 6584 MiB         | 27082 MiB        | Out of Memory    |
> | GPS-Gaussian                          | 1909 MiB         | 2357 MiB         | 4035 MiB         | 11215 MiB        |
> | EVA-Gaussian                          | 2289 MiB         | 3171 MiB         | 7185 MiB         | 24121 MiB        |
>
> --------------------------------------
>
> **Table 2** Comparison of the temporal and computational efficiency among different attention modules.
>
> |  Input Size                                                      ||  2 * 64 * 128 * 128         || |2 * 64 * 256 * 256    || |2 * 32 * 256 * 256   ||
> |---------------------------------------|-----------------------|----------|------------------|-----------------------|----------|------------------|-----------------------|----------|------------------|
> | Module                                | Time (Inference Once) | Params   | GPU Memory Usage | Time (Inference Once) | Params   | GPU Memory Usage | Time (Inference Once) | Params   | GPU Memory Usage |
> | **Self-Attention (in MVSplat)**       | 0.0353s               | 0.789M   | 3808 MiB         | 0.304s                | 0.789M   | 36290 MiB        | 0.263s                | 0.198M   | 32536 MiB        |
> | **Epipolar Attention (in PixelSplat)**| 0.0583s               | 5.062M   | 15554 MiB        | 0.193s                | 5.062M   | 60562 MiB        | 0.169s                | 3.194M   | 59404 MiB        |
> | **EVA Attention (window size=16)**    | 0.007225s             | 0.0661M  | 944 MiB          | 0.0177s               | 0.0661M  | 2200 MiB         | 0.0143s               | 0.0167M  | 1404 MiB         |
> | **EVA Attention (window size=32)**    | 0.006533s             | 0.0661M  | 944 MiB          | 0.0149s               | 0.0661M  | 2192 MiB         | 0.0116s               | 0.0167M  | 1404 MiB         |
> | **EVA Attention (window size=64)**    | 0.006307s             | 0.0661M  | 944 MiB          | 0.0139s               | 0.0661M  | 2192 MiB         | 0.0106s               | 0.0167M  | 1404 MiB         |
> | **EVA Attention (window size=256)**   | 0.006565s             | 0.0661M  | 944 MiB          | 0.0234s               | 0.0661M  | 2192 MiB         | 0.0167s               | 0.0167M  | 1404 MiB         |

---

> > ### Comment · Reviewer_bSU5 · 2024-11-18
> >
> > Thanks for the reply. My concerns have been well solved. This paper extends the setting of GPS-Gaussian to allow the generation by using views across large angles. The running overhead is also acceptable. A potential problem is that the design of EVA assumes all cameras should be at a similar height and with a similar pitch angle, which may limit its generalizability to various scenarios. Nevertheless, this may be feasible in some human capturing scenes with fixed cameras. I'll keep my rating to borderline accept.

---

> > > ### Author Response · Authors · 2024-11-22
> > >
> > > Dear Reviewer bSU5,
> > >
> > > Thank you so much for your insightful feedback and for taking the time to review our work. We greatly appreciate your kind words. Your comments and insights have been invaluable in refining our paper.

---

### Official Review · Reviewer_WLeF · 2024-10-31

**Soundness:** 3
**Presentation:** 3
**Contribution:** 3
**Rating:** 6
**Confidence:** 3

**Summary:**

The paper pays attention on real-time human novel view synthesis with a new pipeline called EVA-Gaussian across diverse camera settings. based on 3D Gaussian representation, which is composed of a position estimation stage, a attributes estimation stage, and a feature refinement stage. To improve position estimation, the paper designs an Efficient cross-View Attention (EVA) module to enhance multi-view correspondence retrieval. Then a recurrent feature refiner is proposed to mitigate geometric artifacts caused by position estimation errors. Experiments on THuman2.0 and THumansit demonstrate the effectiveness of the proposed pipeline.

**Strengths:**

1 A new human novel view synthesis pipeline composed of a position estimation stage, a attributes estimation stage, and a feature refinement stage with 3D Gaussian.

2 An Efficient cross-View Attention module to enhance learning of 3D Gaussian.

3 A recurrent feature refiner that fuses RGB images and feature maps to mitigate geometric artifacts caused by position estimation errors.

**Weaknesses:**

1 The movitivation of EVA shoule be expressed more clearly. The EVA module uses cross-view attention to enhance 3D Gaussian position learning. However, the idea have been used in various feature matching methods to establish correspondences across different veiws, such as SuperGlue [1], LoFTR [2], DKM [3]. It is suggested to explain the module more clearly.

[1] SuperGlue: Learning feature matching with graph neural networks. In CVPR, 2020.

[2] LoFTR: Detector-Free Local Feature Matching with Transformers. In CVPR, 2021.

[3] DKM: Dense Kernelized Feature Matching for Geometry Estimation


2 In Table 1, the paper presents the running time of each method wat a resolution of 256×256. However, GPS-Gaussian declares that it can synthesize 2K-resolution novel views with 25 FPS. This situation is suggeseted to be explained more clearly.

**Questions:**

The questions and suggestions are listed in the part of Weaknesses.

---

> ### Author Response · Authors · 2024-11-17
> **Response 1/3 to Reviewer WLeF(Q1)**
>
> # Dear Reviewer WLeF,
>
> We sincerely thank the reviewer for the constructive comments. We hope the following responses well address the concerns.
>
> **Q.1** **The movitivation of EVA shoule be expressed more clearly. The EVA module uses cross-view attention to enhance 3D Gaussian position learning. However, the idea have been used in various feature matching methods to establish correspondences across different veiws, such as SuperGlue [1], LoFTR [2], DKM [3]. It is suggested to explain the module more clearly.**
>
> **A.1** Thank you for the suggestion. We agree that many feature matching methods utilize cross-view attention to establish correspondences across different views, such as the Attentional Graph Neural Network in SuperGlue [1] and the Local Feature Transformer in LoFTR [2]. To address your concern, we have provided a more detailed explanation in the revised Appendix to clarify the motivation, novelty, and contributions of EVA.
>
> While our EVA module shares the underlying principle of cross-view attention with these approaches, it introduces significant innovations tailored to the specific requirements of our application. For instance, SuperGlue [1] leverages an Attentional Graph Neural Network to selectively focus on relevant keypoint image features, integrating embedded 3D positional information to construct a local graph. However, the attention mechanisms in this framework are restricted to interactions among these keypoints. In contrast, our EVA module employs cross attention more broadly. Although it also focuses on the most relevant pixels, it is designed to output a dense depth map rather than sparse matching results. This allows EVA to encompass pixels across the entire image, enhancing spatial awareness. Similarly, while the Local Feature Transformer in LoFTR [2] performs both self and cross attention on a reduced set of intermediate features, our EVA specifically implements cross attention within a 1D localized window along the x-axis. This design enables EVA to efficiently focus on relevant pixels, while minimizing computational and temporal overhead. The comparison results in Table 1 demonstrate the effectiveness and superiority of our EVA module compared to existing attention mechanisms in feed-forward 3D Gaussian reconstruction.
>
> It is important to note that EVA is specifically tailored to the camera setting used in feed-forward human reconstruction. In this context, where cameras are closely positioned and oriented towards the same point on the human body, the correspondence connections between matched pairs align parallel to the x-axis, as depicted in Figure 8 of the revised Appendix. This specific alignment allows us to simplify traditional cross-view attention mechanisms, like epipolar attention [4]. Unlike existing methods, such as LoFTR [2] and GTA [5], which rely on extensive attention across broader pixel ranges, our approach focuses on nearby pixels along the x-axis within a 1D localized window. Moreover, considering that correspondences may not be perfectly aligned with the x-axis, we implement this attention mechanism within the deeper layers of the UNet architecture, as shown in Figure 9 of the revised Appendix. In these layers, the features of each pixel are aggregated from its neighboring pixels through preceding convolutional layers, thereby enhancing the robustness of feature matching. In addition, to mitigate the potential loss of multiview correspondences at the boundaries of local windows, we perform the attention mechanism twice, with the second iteration using a window shifted by half its size. Figure 10 in the revised Appendix illustrates the key differences between EVA and other attention mechanisms, demonstrating the efficiency gains achieved through our approach.

---

> ### Author Response · Authors · 2024-11-17
> **Response 2/3 to Reviewer WLeF (Q1 Table)**
>
> **Table 1** Comparison of the temporal and computational efficiency among different attention modules.
>
> | Input Size                                                    ||  2 * 64 * 128 * 128         || |2 * 64 * 256 * 256    || |2 * 32 * 256 * 256   ||
> |---------------------------------------|-----------------------|----------|------------------|-----------------------|----------|------------------|-----------------------|----------|------------------|
> | Module                                | Time (Inference Once) | Params   | GPU Memory Usage | Time (Inference Once) | Params   | GPU Memory Usage | Time (Inference Once) | Params   | GPU Memory Usage |
> | **Self-Attention (in MVSplat)**       | 0.0353s               | 0.789M   | 3808 MiB         | 0.304s                | 0.789M   | 36290 MiB        | 0.263s                | 0.198M   | 32536 MiB        |
> | **Epipolar Attention (in PixelSplat)**| 0.0583s               | 5.062M   | 15554 MiB        | 0.193s                | 5.062M   | 60562 MiB        | 0.169s                | 3.194M   | 59404 MiB        |
> | **EVA Attention (window size=16)**    | 0.007225s             | 0.0661M  | 944 MiB          | 0.0177s               | 0.0661M  | 2200 MiB         | 0.0143s               | 0.0167M  | 1404 MiB         |
> | **EVA Attention (window size=32)**    | 0.006533s             | 0.0661M  | 944 MiB          | 0.0149s               | 0.0661M  | 2192 MiB         | 0.0116s               | 0.0167M  | 1404 MiB         |
> | **EVA Attention (window size=64)**    | 0.006307s             | 0.0661M  | 944 MiB          | 0.0139s               | 0.0661M  | 2192 MiB         | 0.0106s               | 0.0167M  | 1404 MiB         |
> | **EVA Attention (window size=256)**   | 0.006565s             | 0.0661M  | 944 MiB          | 0.0234s               | 0.0661M  | 2192 MiB         | 0.0167s               | 0.0167M  | 1404 MiB         |

---

> ### Author Response · Authors · 2024-11-17
> **Response 3/3 to Reviewer WLeF (Q2)**
>
> **Q.2** **In Table 1, the paper presents the running time of each method wat a resolution of 256×256. However, GPS-Gaussian declares that it can synthesize 2K-resolution novel views with 25 FPS. This situation is suggeseted to be explained more clearly.**
>
> **A.2**  Thank you for this comment. In Table 1 of our paper, we present a comprehensive comparison between our proposed EVA-Gaussian method and existing feed-forward 3D Gaussian reconstruction methods, including GPS-Gaussian, PixelSplat, MVSplat, and MVSGaussian. We evaluate these methods using several metrics, PSNR, SSIM, and LPIPS, while also including inference time to assess real-time performance.
>
> It is important to note that the high GPU memory demands of PixelSplat, MVSplat, and MVSGaussian limit their ability to process high-resolution source view images. To ensure a fair comparison, all methods are tested at a low resolution of 256×256. As indicated in Table 1, our EVA-Gaussian method achieves the best performance, with an inference time that is merely 15 ms longer than GPS-Gaussian, but still 15 ms faster than the third fastest method.
>
> While GPS-Gaussian is capable of synthesizing 2K-resolution novel views with an impressive 25 FPS, it does so by using 1024×1024 source images as input. To ensure a fair assessment, in Fig. 1  of our paper, we have compared the reconstruction speeds of EVA-Gaussian and GPS-Gaussian at this higher resolution of 1024×1024, utilizing the same NVIDIA A800 GPU. The results demonstrate that EVA-Gaussian is only 17 ms slower than GPS-Gaussian, while significantly improving the visual quality of the synthesized images.
>
> In addition, we would like to emphasize that both GPS-Gaussian and EVA-Gaussian generate 3D models capable of rendering at 2K resolution. To clarify this point and address any potential confusion, we have added a comparison of EVA-Gaussian and GPS-Gaussian for 2K-resolution novel view synthesis in the revised Appendix.
>
>
> [1] SuperGlue: Learning feature matching with graph neural networks. In CVPR, 2020.
>
> [2] LoFTR: Detector-Free Local Feature Matching with Transformers. In CVPR, 2021.
>
> [3] DKM: Dense Kernelized Feature Matching for Geometry Estimation
>
> [4] He, Y., Yan, R., Fragkiadaki, K., & Yu, S. I. (2020). Epipolar transformers. In Proceedings of the ieee/cvf conference on computer vision and pattern recognition (pp. 7779-7788).
>
> [5] Geiger, Andreas, et al. "GTA: A Geometry-Aware Attention Mechanism for Multi-View Transformers." (2023).

---

> > ### Author Response · Authors · 2024-11-22
> >
> > # Dear Reviewers WLeF,
> >
> > We kindly remind you to review our revisions and individual responses to evaluate if they can address your concerns. If our responses and additional results have sufficiently addressed your concerns, we would greatly appreciate your consideration of increasing your score. We are more than happy to address any remaining questions and concerns. We look forward to hearing from you again.
> >
> > **Best Regards,**
> >
> > The Authors

---

> > > ### Author Response · Authors · 2024-11-25
> > >
> > > Dear Reviewer,
> > >
> > > Thank you for handling our manuscript and providing valuable feedback. We hope that our responses have sufficiently addressed the concerns you raised. We welcome more discussion if you have more questions and suggestions. As the discussion deadline is approaching, we would be very grateful if you could take a moment to review our reply.

---

> > > > ### Comment · Reviewer_WLeF · 2024-11-26
> > > >
> > > > Thanks for the detailed responses. The authors addressed most of my concerns in the rebuttal phase, and thus I would like to raise my score to 6.

---

> > > > > ### Author Response · Authors · 2024-11-26
> > > > >
> > > > > Dear Reviewer WLeF,
> > > > >
> > > > > Thank you so much for your insightful feedback and for taking the time to review our work. We greatly appreciate your kind words. Your comments and insights have been invaluable in refining our paper.

---

### Official Review · Reviewer_qNfu · 2024-11-03

**Soundness:** 3
**Presentation:** 3
**Contribution:** 1
**Rating:** 5
**Confidence:** 5

**Summary:**

The paper proposes a multiview-based method for human novel view synthesis using a 3D Gaussian representation, which consists of three main steps.
First, it estimates position maps with cross-view attention. Next, it combines the position maps and raw images to estimate Gaussian parameters along with feature embedding. Finally, a feature refiner is applied to correct artifacts by splatting features and refining the output image.

**Strengths:**

- The paper is well-written, well-structured, and clear.
- The proposed EVA module is effective in initializing the Gaussian position, outperforming one-step Gaussian parameter regression.
- Incorporating features with original 3D Gaussian and splatting to refine rendering results is interesting.

**Weaknesses:**

- There are concerns about time expansion regarding the project page provided in the abstract.
- Methods like GPS address the sparse-view human Gaussian splatting problem, which conflicts with lines 33–35 of the abstract.
- The claim of "diverse camera settings" is somewhat overstated, as the method cannot resolve single-view settings, which are addressed by other approaches, such as HumanSplat [1].
- The method cannot infer the back view when only front stereo views are provided, as demonstrated in their video.
- The ablation study implies that the effectiveness of anchor loss is limited.
- There are no examples of in-the-wild scenarios presented.
- The video exhibits issues with jumping and transparency.


[1] HumanSplat: Generalizable Single-Image Human Gaussian Splatting with Structure Priors

**Questions:**

- Can the method only interpolate between input views, or is it able to hallucinate invisible parts?
- What causes the jumping and transparency issues in the video?
- How does the shifted window embedding strategy work? Are the window positions for cross-attention consistent across different image sources?
- Does "any" in line 197 include views such as up and side-up perspectives?
- Why do regularizing opacities help ensure consistency? Could it contribute to transparency artifacts, as seen in the video results?
- GPS results show broken hands and feet; why does the proposed method avoid this issue? Since the method does not use human priors, would it also work for objects if the dataset were changed?
- What specific baseline is used for the ablation of the EVA module?
---
I may change my opinion depending on the authors' rebuttal and whether they can address my concerns.

---

> ### Author Response · Authors · 2024-11-17
> **Response 1/5 to Reviewer qNfu (Weakness 1-4)**
>
> # Dear Reviewer qNfu,
>
> We sincerely thank the reviewer for the constructive comments. We hope the following responses well address the concerns.
>
> As for the Weaknesses:
>
> **W.1** **There are concerns about time expansion regarding the project page provided in the abstract.**
>
> **W.A.1** Thank you for raising this concern. We would like to clarify that the most recent update to our project page on our anonymous Github account (https://github.com/anonymousiclr2025) was made on Tue, 01 Oct 2024 03:17:03 GMT. This timestamp aligns with the submission guidelines provided, ensuring that our project page adheres to the required timelines. Therefore, there are no issues related to time expansion.
>
> **W.2** **Methods like GPS address the sparse-view human Gaussian splatting problem, which conflicts with lines 33–35 of the abstract.**
>
> **W.A.2** Thank you for this comment. We acknowledge that our original abstract did not sufficiently clarify the concept of sparse camera settings. Specifically, in accordance with the camera settings in [2], our focus is on addressing sparse-view human Gaussian reconstruction when the camera view angle exceeds 60 degrees. In contrast, GPS-Gaussian employs camera view angles of 45 degrees, which does not fall within the sparse-view category in this context. We have revised the abstract to make this point clearer.
>
> **W.3** **The claim of "diverse camera settings" is somewhat overstated, as the method cannot resolve single-view settings, which are addressed by other approaches, such as HumanSplat [1].**
>
> **W.A.3** Thank you for this comment. We agree that EVA-Gaussian is not designed to effectively handle monocular camera settings due to its cross-view attention design. However, it is applicable to a wide range of multi-camera configurations in human reconstruction. Specifically, EVA-Gaussian effectively accommodates varying camera view angles and different numbers of cameras. As demonstrated in Table 2 of our paper, EVA-Gaussian performs effectively under different camera angles, while Table 3 illustrates its capability with varying numbers of cameras.
>
> It is important to clarify that we did not claim our EVA-Gaussian can resolve all possible camera settings. Instead, our focus is on demonstrating its ability to effectively handle a diverse set of multi-view scenarios.
>
> In addition, Humansplat [1] relies on the human SMPL model, which is difficult to obtain in real-world applications, as discussed in lines 120–129 of our paper. Moreover, it requires 9.3 seconds to generate novel views from source images, significantly hindering its practicality. In contrast, both GPS-Gaussian and our EVA-Gaussian are optimized for real-time human reconstruction tasks, making them more suitable for practical applications. **Therefore, in terms of inference speed and applicability, HumanSplat is not directly comparable to our proposed method.**
>
> To address your concern, we have revised the abstract to clearly emphasize our focus on diverse multi-view camera settings.
>
> **W.4** **The method cannot infer the back view when only front stereo views are provided, as demonstrated in their video.**
>
> **W.A.4** Thank you for your comment. Indeed, incomplete reconstruction is a significant limitation inherent in all feed-forward 3D Gaussian reconstruction methods, **as these approaches primarily focus on reconstruction rather than generation.** Moreover, for certain downstream applications that require precise representations of the human body, such as human-robot interaction [3][4], **it is crucial to avoid hallucinating invisible parts, as this could lead to inaccurate and potentially misleading outcomes.**
>
> In contrast to the previous SOTA method, GPS-Gaussian, our proposed EVA-Gaussian addresses the challenge of limited viewpoint angles by enabling feed-forward 3D reconstruction using source images captured from a broader range of camera angles. Notably, EVA-Gaussian can accommodate four or more input views, provided GPU resources are sufficient, as illustrated in our experimental results. This capability allows EVA-Gaussian to utilize a limited set of front-facing images to infer a comprehensive 3D model that can be rendered from a wider range of camera angles.

---

> ### Author Response · Authors · 2024-11-17
> **Response 2/5 to Reviewer qNfu (Weakness 5-7)**
>
> *W.5** **The ablation study implies that the effectiveness of anchor loss is limited.**
>
> **W.A.5** Thank you for this comment. It is important to note that the anchor loss is specifically designed to regularize the critical facial areas to enhance the overall visual quality of images. Given that facial regions typically constitute only a small fraction of the human body within images, the resulting quantitative performance gains may seem limited. However, as demonstrated by the visualization results presented in our ablation study, the model without anchor loss shows noticeable distortions, particularly in facial regions. The incorporation of anchor loss effectively mitigates these distortions, ensuring that essential facial details are preserved. This underscores the important role of anchor loss in improving human visual perception, even if the improvements in quantitative metrics appear modest.
>
> **W.6** **There are no examples of in-the-wild scenarios presented.**
>
> **W.A.6** Thank you for this comment. We would like to clarify that our primary focus is not on processing casually captured in-the-wild data, such as videos used for reconstructing human avatars. Instead, we concentrate on scenarios where cameras are strategically positioned and images are synchronously captured, for example, in holographic communication systems (see lines 53-54 in the paper). While there are existing systems [6] and in-the-wild datasets from GPS-Gaussian that align with our settings, these datasets are not publicly available.
>
> To address your concern, we have evaluated our model on the publicly available HuMMan dataset, which is a real-world dataset that features a wide camera view angle of 90 degrees, in Figure 13 of the revised Appendix. Notably, GPS-Gaussian fails to generate reasonable results, so we have not included its outcomes. The results demonstrate that EVA-Gaussian can generalize effectively to real-world scenarios.
>
> **W.7** **The video exhibits issues with jumping and transparency.**
>
> **W.A.7** The observed jumping and transparency artifacts arise when novel view cameras approach the boundaries of the 3D model or when there is a significant deviation between the orientations of the novel view cameras and the source view cameras. A more detailed explanation of this phenomenon can be found in the response to your Question 2.
>
> We note that these problems can be effectively mitigated through thoughtful engineering strategies. For instance, increasing the number of source view images used as input can enhance stability. In addition, constraining the novel view cameras to remain within the boundaries defined by the two source view images can further mitigate these issues. We will incorporate these stategies to enhance the visual stability and integrity of the rendered videos on our project page.

---

> ### Author Response · Authors · 2024-11-17
> **Response 3/5 to Reviewer qNfu (Q1-3)**
>
> As for the Questions:
>
> **Q.1** **Can the method only interpolate between input views, or is it able to hallucinate invisible parts?**
>
> **A.1** Thank you for this question. Indeed, our method, as well as all feed-forward 3D Gaussian reconstruction methods, cannot hallucinate invisible parts, **as these approaches primarily focus on reconstruction rather than generation.** Moreover, for certain downstream applications that require precise representations of the human body, such as human-robot interaction [3][4], **it is crucial to avoid hallucinating invisible parts, as this could lead to inaccurate and potentially misleading outcomes.**
>
> However, **this does not mean that our method is simply interpolating between input views**. Our approach first reconstructs the 3D model from multiple source views and then renders them onto a specific image plane for novel view synthesis. 3D reconstruction has long been a fundamental problem in computer vision, and it remains a common practice to estimate 3D models (e.g., point clouds, explicit neural radiance fields, and 3D Gaussians) from multiple images, reproject them into the 3D space, and remap them onto the novel image plane. This procedure heavily relies on the principles of multi-view geometry and **cannot be simply achieved by interpolating between input views**.
>
> In conclusion, our method cannot hallucinate invisible parts and it synthesizes novel views with a strong reliance on multi-view geometry. We hope this addresses your concerns.
>
> **Q.2** **What causes the jumping and transparency issues in the video?**
>
> **A.2** Thank you for this question. In our experimental videos, our EVA-Gaussian method is implemented to ensure a fair comparison with GPS-Gaussian by also utilizing a pair of adjacent source view images for human model inference. This setting inherently restricts the novel view camera angles to lie within the angular range determined by these two source cameras. When the human model rotates beyond this range, additional image pairs from opposing viewpoints are necessary to accurately infer the model and synthesize novel views. This dependence on multiple image pairs leads to transitions between different inferences of the human model, which explains the abrupt transitions observed during smooth view changes. Moreover, the quality of novel view images is primarily guaranteed within the angular boundaries established by the source view cameras. As the novel view camera approaches these boundaries, transparency artifacts emerge due to the limited coverage of the inferred human model.
>
> It is important to note that our EVA-Gaussian is specifically designed to address this issue. EVA-Gaussian can effectively handle scenarios where the angle of source view images is large, allowing novel view cameras to operate in a broader range while maintaining high-quality image synthesis. As shown in Table 2 of our paper, EVA-Gaussian can achieve novel view synthesis with a 90-degree angle, while GPS-Gaussian fails to work effectively. More importantly, EVA-Gaussian is capable of incorporating three or more source view images. This capability ensures consistent human model representation as the novel view camera transitions across intervals spanned by adjacent source cameras, effectively eliminating noticeable artifacts such as transparency and abrupt transitions.
>
> In addition, these issues can be effectively mitigated by engineering designs, such as using more source view images as input or restricting the novel view camera angles to lie within the angular range determined by all source cameras.
>
> **Q.3** **How does the shifted window embedding strategy work? Are the window positions for cross-attention consistent across different image sources?**
>
> **A.3** Thank you for this question. Before carrying out Efficient cross-View Attention, the intermediate features are divided into fixed-length 1D windows along the x-axis, as shown in Fig. 10(C) of the revised Appendix. EVA performs cross-attention between localized windows at identical coordinates across multiple image views, out of the consideration that cameras are closely positioned and oriented towards the same point on the human body in the context of feed-forward human reconstruction. Moreover, considering that correspondences may not be perfectly aligned with the x-axis, we implement this attention mechanism within the deeper layers of the UNet architecture, as shown in Figure 9 of the revised Appendix. In these layers, the features of each pixel are aggregated from its neighboring pixels through preceding convolutional layers, thereby enhancing the robustness of feature matching. In addition, to mitigate the potential loss of multiview correspondences at the boundaries of local windows, we perform the attention mechanism twice, with the second iteration using a window shifted by half its size. This process is demonstrated in Fig. 10(D) of the revised Appendix.

---

> ### Author Response · Authors · 2024-11-17
> **Response 4/5 to Reviewer qNfu (Q4-6)**
>
> **Q.4** **Does "any" in line 197 include views such as up and side-up perspectives?**
>
> **A.4** Thank you for this question. Yes, our model can infer a variety of camera viewpoints, including both up and side-up perspectives. To illustrate this capability, we have included the rendered results in Figure 11 of the revised Appendix.
>
>
> **Q.5** **Why do regularizing opacities help ensure consistency? Could it contribute to transparency artifacts, as seen in the video results?**
>
> **A.5** Thank you for this question. We would like to emphasize that all feed-forward 3D Gaussian reconstruction methods, including GPS-Gaussian, MVSplat, and MVSGaussian, are based on the assumption that the positions of 3D Gaussians directly correspond to the depth maps inferred from source view images. However, this assumption is not theoretically guaranteed.
>
> In our paper, we introduce regularization for both the scales and opacities of the 3D Gaussians to ensure depth consistency. Specifically, as theoretically proved in Appendix C, when the scales of the 3D Gaussians are sufficiently small and their opacities approach either 0 or 1, the rendered depth, which typically represents the depth of the 3D model, aligns precisely with the positions of the 3D Gaussians. This alignment ensures that when we use inferred depth maps to position the 3D Gaussians, the resulting depth of the 3D model remains consistent with the inferred depth maps. Moreover, these inferred depth maps are aligned with the ground truth human model depth during the supervision phase, thereby reinforcing overall depth consistency.
>
> As for the transparency artifacts, it is worth noting that such artifacts are also observed in GPS-Gaussian. This suggests that such artifacts are likely due to discrepancies between the source views and the novel views, rather than the regularization terms themselves. Notably, the opacity regularization term $O_ilog(O_i)$ does not promote transparency. Instead of allowing opacities to settle at intermediate values (e.g., 0.5), this term encourages opacities to converge towards the extremes of 0 or 1, achieving its minimum. Therefore, the regularization effectively minimizes the likelihood of transparency artifacts arising from the opacities of the 3D Gaussians.
>
>
> **Q.6** **GPS results show broken hands and feet; why does the proposed method avoid this issue?**
>
> **A.6** Thank you for this question. GPS-Gaussian works well under dense camera settings, where camera view angles are less than 60 degrees. However, the adaptability of GPS-Gaussian is limited by its reliance on stereo-matching modules. The stereo-matching module first performs stereo rectification, which warps the source view images onto a common image plane, and then searches for correspondencies along the x-axis. Unfortunately, this heavy reliance on stereo rectification can lead to distortion and incompleteness in the rectified images when the camera angles are large. This drawback is reported by GHG [2] and also mentioned in our paper in line 434.
>
> Our proposed method effectively overcomes this limitation by utilizing the EVA module, which removes the reliance on stereo rectification and enables the generation of high-fidelity 3D Gaussian position maps even under significant variations in source camera angles. The effectiveness of the EVA module stems from the integration of strong inductive bias related to camera settings, which significantly reduces both computational load and temporal costs, thereby enhancing the efficiency of 3D Gaussian position estimation.

---

> ### Author Response · Authors · 2024-11-17
> **Response 5/5 to Reviewer qNfu (Q7-8)**
>
> **Q.7** **Since the method does not use human priors, would it also work for objects if the dataset were changed?**
>
> **A.7** Thank you for this question. Since EVA-Gaussian does not rely on explicit human priors, it has the potential to be applied to object-only datasets, provided that the camera priors, e.g., extrinsic and intrinsic parameters, remain consistent. This potential is demonstrated by our cross-dataset validation experiments detailed in the revised Appendix F. Specifically, a model trained on THuman2.0, a dataset exclusively containing human subjects, performs effectively on THumansit, which includes both humans and objects like chairs. This indicates that EVA-Gaussian may generalize well with object datasets.
>
> However, a critical limitation is the absence of a suitable dataset that features fixed and aligned camera parameters to evaluate the performance of EVA-Gaussian when trained on a human dataset. In addition, we currently lack a sufficiently large object dataset to train EVA-Gaussian effectively. Therefore, it remains uncertain to what extent EVA-Gaussian can effectively handle a wide variety of objects.
>
>
> **Q.8** **What specific baseline is used for the ablation of the EVA module?**
>
> **A.8** Thank you for this question. In the ablation of the EVA module, we remove the EVA module from the Gaussian position estimation network $\mathcal{D}^P_{\theta_1}$, which reduces the network to a standard UNet architecture. All other components of the EVA-Gaussian pipeline remain unchanged, allowing us to evaluate the impact of the EVA module on the overall performance.
>
>
> [1] HumanSplat: Generalizable Single-Image Human Gaussian Splatting with Structure Priors
>
> [2] Generalizable Human Gaussians for Sparse View Synthesis. In ECCV, 2024.
>
> [3] Reconstructing human hand pose and configuration using a fixed-base exoskeleton. In ICRA.
>
> [4] Immfusion: Robust mmwave-rgb fusion for 3d human body reconstruction in all weather conditions. In ICRA.
>
> [5] https://web.twindom.com/twinstant-mobile-full-body-3d-scanner/
>
> [6] Volumetric Avatar Reconstruction with Spatio-Temporally Offset RGBD Cameras. In VR 2023

---

> > ### Author Response · Authors · 2024-11-22
> >
> > # Dear Reviewers qNfu,
> >
> > We kindly remind you to review our revisions and individual responses to evaluate if they can address your concerns. If our responses and additional results have sufficiently addressed your concerns, we would greatly appreciate your consideration of increasing your score. We are more than happy to address any remaining questions and concerns. We look forward to hearing from you again.
> >
> > **Best Regards,**
> >
> > The Authors

---

> > > ### Author Response · Authors · 2024-11-25
> > >
> > > Dear Reviewer,
> > >
> > > Thank you for handling our manuscript and providing valuable feedback. We hope that our responses have sufficiently addressed the concerns you raised. We welcome more discussion if you have more questions and suggestions. As the discussion deadline is approaching, we would be very grateful if you could take a moment to review our reply.

---

> > > > ### Comment · Reviewer_qNfu · 2024-11-25
> > > > **Response to rebuttal**
> > > >
> > > > Thank you for your thoughtful rebuttal. Most of my concerns have been addressed, and I could increase my score from 3 to 5. However, I believe that the proposed method for the task does not fully meet the bar for acceptance at ICLR. The task appears to be somewhat rigidly defined. Additionally, it does not seem appropriate to claim that not addressing unseen areas is an advantage through forced explanations.

---

> ### Author Response · Authors · 2024-11-26
>
> # Dear Reviewer qNfu,
>
> Thank you for your feedback and for taking the time to review our work.
>
> We would like to clarify the importance of this task and our contributions to society.
>
> **Recovering novel view images from a set of images captured by well-posed cameras has long been a fundamental task for real-time human novel view synthesis.** Numerous studies, including those based on Signed Distance Fields (SDF) such as PIFu [1], PIFuHD [2], and Function4D [3], as well as methods based on 3D Gaussian splatting like GHG [4], GPS-Gaussian [5], and others [6][7], have aimed to solve this task under well-posed camera settings. Subsequent works have demonstrated that these algorithms can be effectively deployed in real-world systems, such as VirtualCube [8], Tele-Aloha [9], and GPS-Gaussian+ [10], where they perform well in critical applications like holographic communication and human-robot interaction.
>
> Our approach, EVA-Gaussian, operates within the same framework as previous methods, utilizing feed-forward 3D Gaussian reconstruction techniques. Our main contribution is that we are the first to fully leverage the priors inherent in the camera settings. We have designed powerful components, including the EVA module, feature refinement module, and anchor loss, which have proven exceptionally beneficial for the task of human novel view synthesis, whether under dense camera settings (with camera angles less than 60 degrees) or sparse camera settings (with camera angles larger than 60 degrees). Notably, our work consistently outperforms previous methods in terms of the quality of novel view images while maintaining reasonable computational and temporal costs.
>
> In addition, **none of the aforementioned methods, including Function4D [3], Pixelsplat [6], MVSplat [11], MVGaussian [12], GHG [4], GPS-Gaussian [5], VirtualCube [8], Tele-Aloha [9], and GPS-Gaussian+ [10], hallucinate invisible parts**, as all these works aim to recover a realistic human novel view in real-time. This realistic reconstruction can benefit **certain** downstream tasks such as human-robot interaction. We also want to point out that all these methods can reconstruct a complete human model, given a sufficient number of surrounding source view images (for example, a minimum of four surrounding views in EVA-Gaussian).
>
> On the other hand, there are indeed other works that leverage generative models [13][14] or human prior models [15][16] (e.g., SMPL) for human novel view synthesis. We fully acknowledge that these works contribute significantly to human avatar reconstruction, achieving complete 3D human models from limited front views. However, this constitutes a different line of research, often accompanied by high temporal costs (e.g., Humansplat takes 100 times longer than EVA-Gaussian) and may yield unrealistic reconstruction results. Real-time human reconstruction for novel view synthesis emphasizes achieving real-time and realistic reconstruction results and benefits many important downstream tasks such as holographic communication. All these points have been thoroughly discussed in our paper.
>
> Furthermore, we would like to claim that **our EVA-Gaussian is currently the most effective solution for the task of real-time human novel view synthesis.**
>
>
> **Best Regards,**
>
> The Authors
>
> [1] PIFu: Pixel-Aligned Implicit Function for High-Resolution Clothed Human Digitization, in CVPR 2019
>
> [2] PIFuHD: Multi-Level Pixel-Aligned Implicit Function for High-Resolution 3D Human Digitization, in CVPR 2020
>
> [3] Function4D: Real-time Human Volumetric Capture from Very Sparse Consumer RGBD Sensors, in CVPR 2021
>
> [4] Generalizable Human Gaussians for Sparse View Synthesis, in ECCV 2024
>
> [5] GPS-Gaussian: Generalizable Pixel-wise 3D Gaussian Splatting for Real-time Human Novel View Synthesis, in CVPR 2024
>
> [6] Learning to Infer Implicit Surfaces without 3d Supervision, in NeurIPS 2019
>
> [7] Deep Volumetric Video from Very Sparse Multi-view Performance Capture, in ECCV 2018
>
> [8] VirtualCube: An Immersive 3D Video Communication System
>
> [9] Tele-Aloha: A Low-budget and High-authenticity Telepresence System Using Sparse RGB Cameras, in SIGGRAPH 2024
>
> [10] GPS-Gaussian+: Generalizable Pixel-wise 3D Gaussian Splatting for Real-Time Human-Scene Rendering from Sparse Views
>
> [11] MVSplat: Efficient 3D Gaussian Splatting from Sparse Multi-View Images, in ECCV 2024
>
> [12] Fast Generalizable Gaussian Splatting Reconstruction from Multi-View Stereo, in ECCV 2024
>
> [13] HumanSplat: Generalizable Single-Image Human Gaussian Splatting with Structure Priors
>
> [14] StdGEN: Semantic-Decomposed 3D Character Generation from Single Images
>
> [15] Expressive Whole-Body 3D Gaussian Avatar
>
> [16] GaussianAvatar: Towards Realistic Human Avatar Modeling from a Single Video via Animatable 3D Gaussians

---

### Official Review · Reviewer_KDU6 · 2024-11-04

**Soundness:** 3
**Presentation:** 2
**Contribution:** 2
**Rating:** 5
**Confidence:** 4

**Summary:**

This paper proposes a real-time human novel view synthesis framework that leverages 3D Gaussians as the core representation. The approach employs a cross-view attention module to estimate geometric properties for each 3D Gaussian and uses image features to predict their attributes. The novel view image is generated through a splatting technique refined in a recurrent manner.

**Strengths:**

The experimental results demonstrate improved performance over GPS-Gaussian, particularly in the facial region.

**Weaknesses:**

he proposed pipeline shows considerable similarity to GPS-Gaussian. While GPS-Gaussian uses cost volume to estimate depth from two selected views, this work relies on cross-view attention. Furthermore, embedding image features into depth maps is also similar in both approaches.

The novelty and specific contributions of the cross-view attention module are unclear. Cross-view transformers, as implemented here, have been extensively explored in prior work, such as [A] and [B]. A more thorough analysis and additional experiments are needed to compare the proposed EVA module with existing techniques.

The use of facial landmarks as a regularization method is intuitive but not unique. While many related works are focused on human head avatars, this approach is already well-established and should be appropriately credited and compared with existing methods.

Although the paper emphasizes that the proposed approach is real-time and adaptable to various camera settings, this adaptability seems largely due to the two-view correlation strategy, which should also apply to GPS-Gaussian. It is unclear what unique contributions in this work specifically enhance real-time performance and camera adaptability.

The experiment videos show noticeable artifacts, similar to those seen in GPS-Gaussian, including transparent Gaussians in novel views and abrupt transitions during smooth view changes.

[A] Zhou, Brady, and Philipp Krähenbühl. "Cross-view transformers for real-time map-view semantic segmentation." Proceedings of the IEEE/CVF Conference on Computer Vision and Pattern Recognition, 2022.
[B] Pham, Trung X., Zhang Kang, and Chang D. Yoo. "Cross-view Masked Diffusion Transformers for Person Image Synthesis." arXiv preprint arXiv:2402.01516, 2024.

**Questions:**

Given the points raised in the weaknesses, could the authors clarify the main novelty and contribution of this paper?
What specific advantages does the proposed EVA module offer?

---

> ### Author Response · Authors · 2024-11-17
> **Response 1/7 to Reviewer KDU6 (Q1)**
>
> # Dear Reviewer KDU6,
>
> We sincerely thank the reviewer for the constructive comments. We hope the following responses well address the concerns.
>
> **Q.1** **The proposed pipeline shows considerable similarity to GPS-Gaussian. While GPS-Gaussian uses cost volume to estimate depth from two selected views, this work relies on cross-view attention. Furthermore, embedding image features into depth maps is also similar in both approaches.**
>
> **A.1** Thank you for this comment. While our proposed pipeline may initially appear similar to GPS-Gaussian, there are significant differences in both depth estimation and feature embedding.
>
> Traditional cross-volume attention mechanisms estimate depth by constructing a cost volume from multiple selected views. This process is computationally intensive, as it requires computing probability scores across a wide range of hypothesized depth values to form the volume. Consequently, this method becomes increasingly inefficient and less scalable when handling multiple views or higher-resolution images, which limits its effectiveness in more complex scenarios. While GPS-Gaussian adapts the cost volume to facilitate high-resolution processing with reduced temporal and computational resources by leveraging stereo-matching algorithms, it remains heavily dependent on stereo rectification. This reliance poses a significant limitation in sparse camera setups, especially when camera angles exceed 60 degrees.
>
> In contrast, our EVA-Gaussian introduces an Efficient cross-View Attention (EVA) module that streamlines the aggregation of information across multiple views. The EVA module enhances computational efficiency by focusing on relevant features and enables robust depth estimation even under sparse camera settings. This novel attention design not only reduces computational overhead but also improves scalability, making our approach more suitable for complex and high-resolution environments.
>
> We also note that several feed-forward 3D Gaussian reconstruction methods share a similar pipeline that embeds image features into depth maps to infer Gaussian attribute maps, like MVSGaussian [C] and TranSplat [D]. However, the specific procedures for feature embedding and post-processing in our pipeline differ significantly from existing approaches. Most current methods, such as PixelSplat and MVSplat, jointly estimate depth maps and attribute maps, making it difficult to supervise the details of the estimated depth map. Although GPS-Gaussian addresses this issue by separating the estimation of depth maps and attribution maps, it does not extend 3D Gaussian attributes to capture more spatial information for mitigating geometric errors. In contrast, we incorporate a **Feature Refinement** process to encode spatial information in Gaussian feature attributes and enhance the overall image quality in a recurrent manner. In addition, to ensure that 3D Gaussians align accurately with the surface of the human body, we introduce a **3D Gaussian Attribute Regularization** method. This approach maintains the consistency between the depth of the 3D model and the inferred multi-view depth, thereby ensuring multi-view consistency in critical facial areas across different source views.
>
> In summary, our pipeline uniquely integrates **Efficient cross-View Attention**, **Feature Refinement**, and **3D Gaussian Attributes Estimation** to achieve real-time human 3D Gaussian reconstruction.  These innovations not only distinguish our pipeline from GPS-Gaussian but also allow us to achieve SOTA performance across various datasets and camera settings under multiple metrics. A more detailed discussion of the novelty of the proposed pipeline is provided in the revised Appendix D.

---

> ### Author Response · Authors · 2024-11-17
> **Response 2/7 to Reviewer KDU6 (Q2)**
>
> **Q.2** **The novelty and specific contributions of the cross-view attention module are unclear. Cross-view transformers, as implemented here, have been extensively explored in prior work, such as [A] and [B]. A more thorough analysis and additional experiments are needed to compare the proposed EVA module with existing techniques.**
>
> **A.2**  Thank you for this comment. We agree with the reviewer that cross-view transformers have been extensively explored and have demonstrated impressive performance across various applications. However, we would like to clarify that our EVA module is specifically tailored for the task of feed-forward human 3D Gaussian reconstruction, which necessitates reconstructing high-resolution 3D human models from images (e.g., $1024\times1024$) in real time, ideally within 100 milliseconds. Unlike general cross-view transformers, EVA leverages a strong inductive bias specifically designed for the camera settings in this task. By performing cross-view attention exclusively among nearby pixels within a shifted window, EVA achieves localized attention. This approach enhances computational efficiency while preserving essential spatial relationships critical for accurate 3D reconstruction.
>
> The cross-view model [A] referenced by the reviewer is specialized for segmentation tasks and operates between BEV features and RGB image features. In this framework, the BEV feature map serves as the query, while multi-view image features act as the keys and values. In contrast, our EVA performs cross-attention directly between multiple source view image features, enabling a more integrated and seamless fusion of information across views. This fundamental difference allows EVA to better address the requirements of 3D Gaussian reconstruction. Moreover, the cross-view model [B] mentioned is computationally intensive, which requires several seconds for inference even at a lower resolution of $256\times256$. Similarly, large cross-view transformer modules used in 3D generation tasks suffer from significant limitations in speed and GPU memory consumption, making them impractical for our real-time requirements. We have discussed these limitations in our manuscript (lines 145-148) to highlight the inapplicability of such models.
>
> The most closely related attention mechanisms to our work are the epipolar attention used by PixelSplat and the self-attention employed by MVSplat. As shown in Table 1, we have conducted a comparative analysis of our module against these approaches, focusing on both temporal and computational costs. We have also adapted the attention mechanism from [A] to our task to ensure a fair comparison. The results in Table 1 demonstrate the effectiveness and superiority of our EVA module compared to these existing attention mechanisms.
>
> To address your concerns, we have clarified the novelty and contributions of the EVA module in the revised manuscript, including the comparison results in the supplementary material.

---

> ### Author Response · Authors · 2024-11-17
> **Response 3/7 to Reviewer KDU6 (Q2 Table)**
>
> **Table 1** Comparison of the temporal and computational efficiency among different attention modules.
>
> | Input Size                                                    ||  2 * 64 * 128 * 128         || |2 * 64 * 256 * 256    || |2 * 32 * 256 * 256   ||
> |---------------------------------------|-----------------------|----------|------------------|-----------------------|----------|------------------|-----------------------|----------|------------------|
> | Module                                | Time (Inference Once) | Params   | GPU Memory Usage | Time (Inference Once) | Params   | GPU Memory Usage | Time (Inference Once) | Params   | GPU Memory Usage |
> | **Cross Attention (in [A])**          | 0.159s                | 0.0336M  | 49736 MiB        |                    |Out Of Memory|                  |                    |Out Of Memory|                  |
> | **Self-Attention (in MVSplat)**       | 0.0353s               | 0.789M   | 3808 MiB         | 0.304s                | 0.789M   | 36290 MiB        | 0.263s                | 0.198M   | 32536 MiB        |
> | **Epipolar Attention (in PixelSplat)**| 0.0583s               | 5.062M   | 15554 MiB        | 0.193s                | 5.062M   | 60562 MiB        | 0.169s                | 3.194M   | 59404 MiB        |
> | **EVA Attention (window size=16)**    | 0.007225s             | 0.0661M  | 944 MiB          | 0.0177s               | 0.0661M  | 2200 MiB         | 0.0143s               | 0.0167M  | 1404 MiB         |
> | **EVA Attention (window size=32)**    | 0.006533s             | 0.0661M  | 944 MiB          | 0.0149s               | 0.0661M  | 2192 MiB         | 0.0116s               | 0.0167M  | 1404 MiB         |
> | **EVA Attention (window size=64)**    | 0.006307s             | 0.0661M  | 944 MiB          | 0.0139s               | 0.0661M  | 2192 MiB         | 0.0106s               | 0.0167M  | 1404 MiB         |
> | **EVA Attention (window size=256)**   | 0.006565s             | 0.0661M  | 944 MiB          | 0.0234s               | 0.0661M  | 2192 MiB         | 0.0167s               | 0.0167M  | 1404 MiB         |

---

> ### Author Response · Authors · 2024-11-17
> **Response 4/7 to Reviewer KDU6 (Q3)**
>
> **Q.3** **The use of facial landmarks as a regularization method is intuitive but not unique. While many related works are focused on human head avatars, this approach is already well-established and should be appropriately credited and compared with existing methods.**
>
> **A.3**  Thank you for this comment. We agree that the use of facial landmarks as a regularization method is a well-established method for human head avatars. For instance, previous studies such as [E] have employed facial landmarks as a regularization term to effectively control the movement of existing head avatars, while [F] has incorporated facial landmarks as anchors in a triplet loss framework to ensure that the remapping results align with human expressions.
>
> Nonetheless, our approach differs significantly from these methods, which primarily focus on facial expression transfer. Instead, we integrate facial landmarks to regularize the 3D Gaussian attributes of human models, thereby enabling a more consistent and robust 3D reconstruction across multiple viewpoints. Although there are existing works, such as [G], that explore multi-view consistency to enhance model performance, their approaches are based on point cloud multi-view matching. This differs significantly from our 3D Gaussian regularization, as the attributes associated with 3D Gaussian representations, such as scales, opacities, and positions, are fundamentally different from those of point clouds. Importantly, our proposed anchor loss is specifically designed to regularize these 3D Gaussian attributes. As elaborated in Appendix C, we provide a theoretical analysis to demonstrate how this regularization facilitates depth alignment and significantly enhances the quality of novel view synthesis. To the best of our knowledge, our work is the first to apply facial landmark regularization to 3D Gaussian attributes in this context, thereby advancing the state of the art in 3D human reconstruction across multiple views.

---

> ### Author Response · Authors · 2024-11-17
> **Response 5/7 to Reviewer KDU6 (Q4)**
>
> **Q.4** **Although the paper emphasizes that the proposed approach is real-time and adaptable to various camera settings, this adaptability seems largely due to the two-view correlation strategy, which should also apply to GPS-Gaussian. It is unclear what unique contributions in this work specifically enhance real-time performance and camera adaptability.**
>
> **A.4**  Thank you for this comment. We acknowledge that GPS-Gaussian is indeed a real-time method and adaptable to other camera settings. However, the adaptability of GPS-Gaussian is limited by its reliance on stereo-matching modules. The stereo-matching module first performs stereo rectification, which warps the source view images onto a common image plane, and then searches for correspondencies along the x-axis. Unfortunately, this heavy reliance on stereo rectification can lead to distortion and incompleteness in the rectified images when the camera angles are large. Consequently, when the camera angles exceed 60 degrees, GPS-Gaussian struggles to achieve satisfactory performance, as illustrated by the qualitative visualization results presented in the paper. In contrast, our proposed method effectively overcomes this limitation by utilizing the EVA module, which enables the generation of high-fidelity 3D Gaussian position maps even under significant variations in source camera angles. The effectiveness of the EVA module stems from the integration of strong inductive bias related to camera settings, which significantly reduces both computational load and temporal costs, thereby enhancing the efficiency of 3D Gaussian position estimation. To clarify the unique contributions of the EVA module, we provide a detailed explanation in the following:
>
> For multiview correspondence retrieval across different image views, epipolar attention [H] has demonstrated its effectiveness by performing attention along the epipolar lines. This approach is based on the principle that a pixel in the source image corresponds to a pixel along the epipolar line in the target image. However, the sampling and attention processes in traditional epipolar attention are computationally and temporally intensive, as shown in Table 1.
>
> In the context of feed-forward human reconstruction, where cameras are closely positioned and oriented towards the same point on the human body, the correspondence connections between matched pairs align parallel to the x-axis, as depicted in Figure 8 of the revised Appendix. This specific alignment allows us to simplify traditional epipolar attention. Unlike existing methods, such as [A], [I], and [J], which rely on extensive attention across broader pixel ranges, our approach focuses on nearby pixels along the x-axis within a 1D localized window. Moreover, considering that correspondences may not be perfectly aligned with the x-axis, we implement this attention mechanism within the deeper layers of the UNet architecture, as shown in Figure 9 of the revised Appendix. In these layers, the features of each pixel are aggregated from its neighboring pixels through preceding convolutional layers, thereby enhancing the robustness of feature matching. In addition, to mitigate the potential loss of multiview correspondences at the boundaries of local windows, we perform the attention mechanism twice, with the second iteration using a window shifted by half its size. Figure 10 in the revised Appendix illustrates the key differences between EVA and other attention mechanisms, demonstrating the efficiency gains achieved through our approach.
>
> In summary, by simplifying the attention process and exploiting the specific camera configuration inductive biases, EVA-Gaussian not only enhances real-time performance but also significantly improves adaptability to diverse camera settings, surpassing the capabilities of existing methods like GPS-Gaussian.

---

> ### Author Response · Authors · 2024-11-17
> **Response 6/7 to Reviewer KDU6 (Q5)**
>
> **Q.5** **The experiment videos show noticeable artifacts, similar to those seen in GPS-Gaussian, including transparent Gaussians in novel views and abrupt transitions during smooth view changes.**
>
> **A.5** In our experimental videos, our EVA-Gaussian method is implemented to ensure a fair comparison with GPS-Gaussian by also utilizing a pair of adjacent source view images for human model inference. This setting inherently restricts the novel view camera angles to lie within the angular range determined by these two source cameras. When the human model rotates beyond this range, additional image pairs from opposing viewpoints are necessary to accurately infer the model and synthesize novel views. This dependence on multiple image pairs leads to transitions between different inferences of the human model, which explains the abrupt transitions observed during smooth view changes. Moreover, the quality of novel view images is primarily guaranteed within the angular boundaries established by the source view cameras. As the novel view camera approaches these boundaries, transparency artifacts emerge due to the limited coverage of the inferred human model.
>
> It is important to note that our EVA-Gaussian is specifically designed to address this issue. EVA-Gaussian can effectively handle scenarios where the angle of source view images is large, allowing novel view cameras to operate in a broader range while maintaining high-quality image synthesis. As shown in Table 2 of our paper, EVA-Gaussian can achieve novel view synthesis with a 90-degree angle, while GPSGaussian fails to work effectively. More importantly, EVA-Gaussian is capable of incorporating three or more source view images. This capability ensures consistent human model representation as the novel view camera transitions across intervals spanned by adjacent source cameras, effectively eliminating noticeable artifacts such as transparency and abrupt transitions.

---

> ### Author Response · Authors · 2024-11-17
> **Response 7/7 to Reviewer KDU6 (Summary)**
>
> **Q** **Given the points raised in the weaknesses, could the authors clarify the main novelty and contribution of this paper? What specific advantages does the proposed EVA module offer?**
>
> **Summary** Thank you for your questions. We have addressed each point raised in the reviews in detail. To summarize, our paper introduces a novel pipeline for real-time human 3D Gaussian reconstruction that significantly differs from existing approaches like GPS-Gaussian. Our method incorporates several key innovations:
>
> 1. **Efficient Cross-View Attention (EVA) Module**: Unlike traditional cross-volume attention mechanisms, EVA streamlines information aggregation across multiple views by focusing on relevant features within localized windows. This design enhances computational efficiency and scalability, enabling robust depth estimation even with sparse camera setups.
>
> 2. **Feature Refinement**: We have implemented a feature refinement process to better encode spatial information in Gaussian feature attributes. Through recurrent processing, this refinement significantly improves image quality, ensuring that spatial details are meticulously captured and accurately represented in the reconstructed 3D model.
>
> 3. **3D Gaussian Attribute Regularization**: To maintain consistency and fidelity in the reconstructed human models, we incorporate facial landmarks into the regularization of the 3D Gaussian attributes. This integration ensures that the depth of the 3D model aligns with multi-view depth, particularly in critical facial regions, thereby preserving multi-view consistency and enhancing reconstruction accuracy.
>
> With these novel designs, our pipeline not only enhances real-time performance but also significantly improves adaptability to diverse camera settings, outperforming GPS-Gaussian and achieving SOTA performance.
>
> The EVA module offers several advantages over existing attention mechanisms:
>
> - **Computational Efficiency**: As demonstrated in Table 1, EVA significantly reduces inference time and GPU memory usage compared to cross attention [A], self-attention [MVSplat], and epipolar attention [PixelSplat]. Notably, with a window size of 64, EVA achieves an impressive inference time of approximately 0.006 seconds, which is much faster than other methods.
>
> - **Scalability and Adaptability**: EVA is designed for high-resolution 3D human reconstruction. Its localized attention approach allows it to adapt seamlessly to various camera settings without the heavy computational burdens that limit other cross-view transformers.
>
> - **Enhanced Depth Estimation**: By leveraging a strong inductive bias specific to our task, EVA performs cross-view attention among nearby pixels within shifted windows. This strategy preserves essential spatial relationships, leading to more accurate and reliable depth estimations even under challenging conditions with significant camera angle variations.
>
> [A] Zhou, Brady, and Philipp Krähenbühl. "Cross-view transformers for real-time map-view semantic segmentation." Proceedings of the IEEE/CVF Conference on Computer Vision and Pattern Recognition, 2022.
>
> [B] Pham, Trung X., Zhang Kang, and Chang D. Yoo. "Cross-view Masked Diffusion Transformers for Person Image Synthesis." arXiv preprint arXiv:2402.01516, 2024.
>
> [C] Liu, T., Wang, G., Hu, S., Shen, L., Ye, X., Zang, Y., ... & Liu, Z. (2024). Fast Generalizable Gaussian Splatting Reconstruction from Multi-View Stereo. arXiv preprint arXiv:2405.12218.
>
> [D] Zhang, C., Zou, Y., Li, Z., Yi, M., & Wang, H. (2024). Transplat: Generalizable 3d gaussian splatting from sparse multi-view images with transformers. arXiv preprint arXiv:2408.13770.
>
> [E] Onizuka, H., Thomas, D., Uchiyama, H., & Taniguchi, R. I. (2019). Landmark-guided deformation transfer of template facial expressions for automatic generation of avatar blendshapes. In Proceedings of the IEEE/CVF International Conference on Computer Vision Workshops (pp. 0-0).
>
> [F] Liu, C., Ham, J., Postma, E., Midden, C., Joosten, B., & Goudbeek, M. (2013). Representing affective facial expressions for robots and embodied conversational agents by facial landmarks. International Journal of Social Robotics, 5, 619-626.
>
> [G] Luo, X., Huang, J. B., Szeliski, R., Matzen, K., & Kopf, J. (2020). Consistent video depth estimation. ACM Transactions on Graphics (ToG), 39(4), 71-1.
>
> [H] He, Y., Yan, R., Fragkiadaki, K., & Yu, S. I. (2020). Epipolar transformers. In Proceedings of the ieee/cvf conference on computer vision and pattern recognition (pp. 7779-7788).
>
> [I] Geiger, Andreas, et al. "GTA: A Geometry-Aware Attention Mechanism for Multi-View Transformers." (2023).
>
> [J] Wang, X., Zhu, Z., Huang, G., Qin, F., Ye, Y., He, Y., ... & Wang, X. (2022, October). Mvster: Epipolar transformer for efficient multi-view stereo. In European Conference on Computer Vision (pp. 573-591). Cham: Springer Nature Switzerland.

---

> > ### Comment · Reviewer_KDU6 · 2024-11-20
> >
> > Thank you for the detailed feedback and the extensive additional experiments.
> >
> > After carefully reviewing the rebuttal and considering the comments from other reviewers, I still view the Efficient Cross-View Attention Module, Feature Refinement, and Attribute Regularization as incremental contributions that provide limited new insights into the problem. While these components enhance an existing pipeline, they do not address the underlying fundamental challenges. However, I am impressed by the efficiency of the EVA module. I remain inclined toward a negative rating but will not strongly oppose acceptance if there is substantial support from the other reviewers.

---

> > > ### Author Response · Authors · 2024-11-26
> > >
> > > # Dear Reviewer KDU6,
> > >
> > > Thank you once again for your insightful feedback and for taking the time to review our work.
> > >
> > > Finally, we would like to clarify the importance of this task and our contributions to society.
> > >
> > > Recovering novel view images from a set of images captured by well-posed cameras has long been a fundamental task for real-time human novel view synthesis. Numerous studies, including those based on Signed Distance Fields (SDF) such as PIFu [A], PIFuHD [B], and Function4D [C], as well as methods based on 3D Gaussian splatting like GHG [D], GPS-Gaussian [E], and others [F][G], have aimed to solve this task under well-posed camera settings. Subsequent works have demonstrated that these algorithms can be effectively deployed in real-world systems, such as VirtualCube [H], Tele-Aloha [I], and GPS-Gaussian+ [J], where they perform well in critical applications like holographic communication and human-robot interaction.
> > >
> > > Our work utilizes the same framework as these previous methods and employs a feed-forward 3D Gaussian reconstruction pipeline similar to other works such as Pxielsplat, MVSplat, MVGaussian, and GPS-Gaussian. Although these works may appear similar at a high level, each has its unique designs in the underlying model architecture. Our EVA-Gaussian method fully leverages the priors inherent in the camera settings and introduces powerful components, including the EVA module, feature refinement module, and anchor loss. These innovations have proven exceptionally beneficial for the task of human novel view synthesis, whether under dense camera settings (with camera angles less than 60 degrees) or sparse camera settings (with camera angles larger than 60 degrees). Notably, our work consistently outperforms previous methods in terms of the quality of novel view images while maintaining reasonable computational and temporal costs.
> > >
> > > Moreover, we would like to claim that **our EVA-Gaussian is currently the most effective solution for the task of real-time human novel view synthesis.**
> > >
> > >
> > > **Best Regards,**
> > >
> > > The Authors
> > >
> > > [A] PIFu: Pixel-Aligned Implicit Function for High-Resolution Clothed Human Digitization, in CVPR 2019
> > >
> > > [B] PIFuHD: Multi-Level Pixel-Aligned Implicit Function for High-Resolution 3D Human Digitization, in CVPR 2020
> > >
> > > [C] Function4D: Real-time Human Volumetric Capture from Very Sparse Consumer RGBD Sensors, in CVPR 2021
> > >
> > > [D] Generalizable Human Gaussians for Sparse View Synthesis, in ECCV 2024
> > >
> > > [E] GPS-Gaussian: Generalizable Pixel-wise 3D Gaussian Splatting for Real-time Human Novel View Synthesis, in CVPR 2024
> > >
> > > [F] Learning to Infer Implicit Surfaces without 3d Supervision, in NeurIPS 2019
> > >
> > > [G] Deep Volumetric Video from Very Sparse Multi-view Performance Capture, in ECCV 2018
> > >
> > > [H] VirtualCube: An Immersive 3D Video Communication System
> > >
> > > [I] Tele-Aloha: A Low-budget and High-authenticity Telepresence System Using Sparse RGB Cameras, in SIGGRAPH 2024
> > >
> > > [J] GPS-Gaussian+: Generalizable Pixel-wise 3D Gaussian Splatting for Real-Time Human-Scene Rendering from Sparse Views

---

> ### Author Response · Authors · 2024-11-22
>
> Dear Reviewer KDU6,
>
> Thank you so much for your insightful feedback and for taking the time to review our work. We greatly appreciate your kind words and the updated score. Your comments and insights have been invaluable in refining our paper.

---

### Author Response · Authors · 2024-11-17
**General Response**

# Dear Reviewers, ACs, and PCs,

We sincerely thank you for your dedication, support, and insightful feedback on our paper. Your constructive comments have greatly assisted us in enhancing the quality of our paper. We have carefully considered all the feedback, addressed each question, and included new experimental results to strengthen our study. Below, we summarize the main revisions we have made:

## Details on the EVA Module

- [KDU6] [WLeF] We have added a **comparison between EVA and existing attention mechanisms** in Appendix D to clarify its contributions and advantages;
- [KDU6] [qNfu] We have included **more structure and operational details** in Appendix D to provide a clearer understanding of the EVA module's functionality.

## More Experimental Results

- [KDU6] [bSU5] A comprehensive **comparison of the computational cost** among our method, GPS-Gassuain, and existing attention mechanisms has been added in Appendix D to highlight efficiency gains;
- [qNfu] [WLeF] We have included **visualization** results from novel view cameras at 1) **random position** and 2) **higher resolution** in Appendix E to demonstrate the effectiveness of our method in diverse scenarios;
- [bSU5] We have incorporated **cross-dataset validation** results in Appendix F to validate the robustness of our method across different datasets;
- [bSU5] An evaluation using **real-world data** has been added in Appendix G to further validate our approach in practical applications.

## Improvements in Writing and Presentation

- [qNfu] We have included Humansplat in comparison in Sec. 2;
- [qNfu] The term "sparse camera setting" has been clearly defined in the abstract to ensure clarity;
- [qNfu] The phrase "diverse camera setting" has been revised to "diverse multi-view camera setting" in the abstract for improved specificity;
- [bSU5] Typos in Figure 2 have been corrected.

We kindly invite the reviewers to examine our revisions and detailed responses. We are more than willing to address any remaining questions or concerns. If our responses and additional results sufficiently address your concerns, we would greatly appreciate your consideration of increasing your scores. We are truly grateful for your engagement and invaluable suggestions, which have significantly contributed to enhancing our work.


**Best Regards,**

The Authors

---

### Meta-Review · Area_Chair_6Ehb · 2024-12-23

**Metareview:**

The paper presents a method for real-time novel view synthesis for humans from multi-view inputs, using 3D Gaussian Splatting. It relies on a cross-view attention module to estimate position for the 3D Gaussians, image features for attribute estimation and a recurrent refinement step. The paper is well-written and presents improved empirical results compared to recent state-of-the-art. However, the methodological choices like cross-view attention and refinement are known in prior works and while their combination is shown to be effective, the obtained abilities are not significantly superior to prior works like GPS-Gaussian. Thus, it is recommended that the paper is not ready for acceptance at ICLR.

**Additional Comments On Reviewer Discussion:**

While the initial scores were weaker, the reviewers considered the author rebuttal, following which two of the reviewers raised their scores. KDU6 remains unconvinced on the contributions with respect to GPS-Gaussian. While qNfu raised their score after the rebuttal, it still leans towards rejection based on the scope of the problem and quality of results. WLeF requires clarifications and leans to accept once they are provided by the rebuttal. Reviewer bSU5 is the most positive after concerns on cross-domain generalization are addressed by the rebuttal, keeping their score to borderline accept. However, despite the author rebuttal, there was no strong support for acceptance among the reviewers.

---

### Decision · Program_Chairs · 2025-01-22

Reject